

# Experimental study on short-text clustering using transformer-based semantic similarity measure

Khaled Abdalgader[1], Atheer A. Matroud[2] and Khaled Hossin[3]

[1] Department of Computer Science and Engineering, American University of Ras Al Khaimah, Ras Al Khaimah, United Arab Emirates
[2] De Montfort University-Dubai, Dubai, United Arab Emirates
[3] Department of Mechanical and Industrial Engineering, American University of Ras Al Khaimah, Ras Al Khaimah, United Arab Emirates

## ABSTRACT

Sentence clustering plays a central role in various text-processing activities and has received extensive attention for measuring semantic similarity between compared sentences. However, relatively little focus has been placed on evaluating clustering performance using available similarity measures that adopt low-dimensional continuous representations. Such representations are crucial in domains like sentence clustering, where traditional word co-occurrence representations often achieve poor results when clustering semantically similar sentences that share no common words. This article presents a new implementation that incorporates a sentence similarity measure based on the notion of embedding representation for evaluating the performance of three types of text clustering methods: partitional clustering, hierarchical clustering, and fuzzy clustering, on standard textual datasets. This measure derives its semantic information from pre-training models designed to simulate human knowledge about words in natural language. The article also compares the performance of the used similarity measure by training it on two state-of-the-art pre-training models to investigate which yields better results. We argue that the superior performance of the selected clustering methods stems from their more effective use of the semantic information offered by this embedding-based similarity measure. Furthermore, we use hierarchical clustering, the best-performing method, for a text summarization task and report the results. The implementation in this article demonstrates that incorporating the sentence embedding measure leads to significantly improved performance in both text clustering and text summarization tasks.

# INTRODUCTION

Although clustering has a long history in natural language processing and computational linguistics literature, many clustering applications on textual data have been applied at the document level. Recently, however, there has been an increased interest in applying clustering techniques to sentence-level text. This has been mostly in response to emerging

Corresponding author
Khaled Abdalgader,
khaled.balawafi@aurak.ac.ae

text-processing activities such as text summarization (*Mojrian & Mirroshandel, 2021*; *Sanchez-Gomez, Vega-Rodríguez & Pérez, 2021*; *Lamsiya et al., 2021*; *Mohd, Jan & Shah, 2020*; *Mutlu, Sezer & Akcayol, 2019*; *Jain, Borah & Biswas, 2023*), text mining (*Yong et al., 2019*; *Choi, Jeon & Kim, 2021*), and sentiment analysis (*Behera et al., 2021*; *Chauhan, Sharma & Sikka, 2021*; *Öztürk & Ayvaz, 2018*; *Abdalgader & Al Shibli, 2020*) which have innovative as being important, especially since the growth of the digital textual collections. One approach, for example, to text summarization is to utilize the main topics which characterize a target document, and then construct an abstractive or an extractive summary of the document by appending a coherent description of each topic. Presumably, sentences related to each other are more likely to belong to the same topic or cluster than sentences unrelated. Therefore, the clustering method that uses an appropriate sentence similarity measure should provide a useful tool for better utilization of those topics.

With respect to the sentence clustering task, it consists of two main parts; the first part is to calculate the semantic similarity between sentences and then cluster them according to their semantic similarity scores. Clustering sentence-level text poses a unique challenge that arises from the fact that sentences typically contain fewer words, making it more challenging to determine semantic similarity between them when compared to clustering at the document level (*Skabar & Abdalgader, 2013*; *Ahmed et al., 2023*). The majority of unsupervised sentence clustering methods, such as those proposed by *Xu et al. (2015)*, *Xu et al. (2017)*, *Patibandla & Veeranjaneyulu (2018)* and *Hadifar et al. (2019)* rely on semantic similarity measures that exclusively use word co-occurrence representations (*e.g.*, one-hot encoded vectors). However, these measures, commonly employed at the document level, are not well-suited for sentence-level clustering because two sentences may share a similar topic even if they do not share any common words. While the assumption that semantic similarity can be measured in terms of word co-occurrence may be valid at the document level, the assumption does not hold for small-sized text fragments such as sentences, since two sentences may be semantically related despite having few, if any, words in common. Therefore, at the sentence level, we require a text representation scheme that is better able to capture the semantic content of sentences, thus enabling more appropriate similarity measures to be defined.

To solve this problem, a wide variety of sentence similarity measures have been presented in the literature (*Kiros et al., 2015*; *Hill, Cho & Korhonen, 2016*; *Conneau et al., 2017*; *Cer et al., 2018*; *Nils & Iryna, 2019*; *Abdalgader & Skabar, 2010*; *Yang et al., 2019*; *Raffel et al., 2020*). Rather than representing sentences in a common vector space (*i.e.,* high-dimensional vector representation), these measures define sentence similarity as some function of inter-sentence word-to-word similarities (*i.e.,* low-dimension continuous representation), where these similarities are in turn usually derived either from distributional information from some corpora (*Islam & Inkpen, 2008*), or semantic information represented in external lexical sources (*Li et al., 2006*; *Mihalcea, Corley & Strapparava, 2006*; *Fellbaum, 1998*). The similarity measures employed in these methods utilize a text representation scheme that involves training distributed word vectors using neural networks. This approach allows them to capture hidden semantic and syntactic patterns within words, which can be leveraged to create similarity measures for words. Consequently, this process facilitates the

calculation of semantic similarity between pairs of sentences using established methods like cosine distance. These similarity measures are commonly referred to as ''sentence embeddings''.

Despite the popularity of sentence embedding measures in computational linguistics, there is an absence of studies that have demonstrated the evaluation results of sentence-level text clustering methods using this type of semantic similarity measure. In other words, the combinations between unsupervised clustering methods and the sentence embedding techniques were not extensively studied (*e.g.*, with few exceptions). The research described in this article, however, is motivated by the belief that the performance of such clustering could be improved by incorporating sentence embedding techniques. The contributions of this article are as follows. First, to the best of our knowledge, we present the first attempt to conduct an experimental and evaluation methodology that allows the comparison of various unsupervised clustering methods using sentence embedding techniques that are trained on the state-of-the-art pre-training models to cluster collections of sentence-level texts. Therefore, the main novelty of this article lies in the use of word embedding together to be applied to deep learning classification using transformers, which is the latest state-of-the-art, with clustering. In our experiments, we compare and evaluate the performance of three types of clustering approaches: partitional clustering (*Wagstaff et al., 2001*; *Ng, Jordan & Weiss, 2001*; *Frey & Dueck, 2007*), hierarchical clustering (*Guo, Zhao & Liu, 2019*; *Roux, 2018*), and fuzzy clustering (*Skabar & Abdalgader, 2013*; *Hathaway, Devenport & Bezdek, 1989*) using sentence similarity measure called SentenceBERT (*Nils & Iryna, 2019*), which based on the vector reduction (*i.e.,* embeddings) as terms for the text representation. The effects of these methods are examined in several experiments that train the used measure with two pre-training models to explore its capability to capture more accurate semantic information. Second, we report extensive experiments on large-scale benchmark datasets to demonstrate the effectiveness of using sentence embedding similarity measures within short text clustering tasks. Finally, we report results obtained by applying the best-performed clustering method to a text summarization task. The implementation results demonstrated that incorporating sentence embedding techniques leads to favourable performance among text clustering and text summarization tasks.

The remainder of the article is organized in the following manner. 'Related Work' describes relevant work in text representation, and both sentence similarity and clustering methods. 'Preliminaries' presents preliminaries to sentence clustering task and 'Experimental Methodology' describes the experimental methodology. Empirical results are presented in 'Empirical Results', and 'Conclusion' concludes the article.

## RELATED WORK

Tasks typically performed in sentence clustering are interdependent. The performance of any clustering method depends on several factors, such as the quality of the input data, the text representation, and the similarity measures used. The performance sentence clustering entirely depends on the quality of the used sentence similarity measure, which in turn depends on the text representation scheme that is better able to capture the semantic

content of sentences. Since our primary interest is in evaluating the clustering performance using sentence embeddings, we concern ourselves in this section with relevant works to the text representation, available sentence embedding similarity measures, and three types of text clustering methods.

## Text representation scheme

The vector space model (*Salton, 1989*) is the most used scheme to represent textual documents as a point in a high-dimensional input space in which each dimension corresponds to a unique word. That is, a document $d_j$ is represented as a vector $\mathbf{x}_j = (w_{1j}, w_{2j}, w_{3j}, \dots)$, where $w_{ij}$ is a weight that represents in some way the importance of word $w_i$ in $d_j$, and is based, at least in part, on the frequency of occurrence of $w_i$ in $d_j$ (term frequency). The similarity between two documents is then calculated using the cosine of the angle between the two vectors. However, this is not the case for sentence-level text, since two sentences may be semantically very similar while containing no common words. Therefore, word co-occurrence (*i.e.,* bag-of-word or one-hot encoding) at the sentence level may be rare or even absent and arises because the characteristic flexibility of natural language enables humans to express similar meanings using quite different sentences in terms of structure and length. Therefore, we require a representation that can better capture the fine-grained semantic content of sentences at the sentence level, thus enabling a more appropriate similarity measure to be defined.

To alleviate the above-described issue, another text representation scheme known as word embeddings (*Mikolov et al., 2013b*; *Mikolov et al., 2013a*; *Pennington, Socher & Manning, 2014*; *McCann et al., 2017*; *Peters et al., 2018*) is currently being developed to reduce high-dimension space (*i.e.,* one-hot encoding) in small-sized vectors. This representation scheme can capture the semantic and syntactic meaning of each word in the corpus vocabulary based on the usage of these words in sentences. Words with similar semantic and contextual meanings also have similar vector representations, while at the same time, each word in the vocabulary will have a unique set of vector representations. This means that words, for example, such as "money" or "currency" will probably have a similar embedding, different from the embedding of the word "water".

The word embedding methods commonly proposed in the literature are known as Word2Vec (*Mikolov et al., 2013b*), GloVe (*Pennington, Socher & Manning, 2014*), and FastText (*Bojanowski et al., 2017*). Word2Vec excels in capturing semantic relationships between words by learning from local context windows, making it effective for tasks reliant on semantic similarity. Conversely, GloVe's emphasis on global word co-occurrence statistics across the entire corpus provides a robust understanding of word relationships, yielding embeddings proficient in capturing broader semantic information. FastText, distinctively, incorporates sub-word information, enabling it to handle out-of-vocabulary words and morphologically rich languages adeptly. Its sub-word-based approach enhances representations of rare words and morphological variants. Despite their successes, these methods present trade-offs: Word2Vec might struggle with rare words, GloVe may overlook morphological intricacies, while FastText's extended processing may impact computational efficiency. Therefore, the selection among these techniques should consider the specific

nuances of the dataset and the application requirements, balancing computational complexity, representation robustness, and the nature of semantic relationships crucial to the sentences similarity and clustering tasks.

## Sentence embedding measures

Various approaches to sentence embeddings have been proposed in recent years (*Hill, Cho & Korhonen*; *Conneau et al., 2017*; *Cer et al., 2018*; *Nils & Iryna, 2019*; *Pei et al., 2020*; *Giorgi et al., 2021*; *Wang & Kuo, 2020*; *Pragst, Minkera & Ultes, 2020*; *Perone, Silveira & Paula, 2018*) to address the restrictions of utilizing multi-word information from word embeddings. One approach to sentence embedding, however, is proposed by *Kiros et al. (2015)*, and is known as SkipThought, which trains an encoder–decoder architecture to predict the surrounding sentences in the given context. Another sentence embedding approach introduced by *Conneau et al. (2017)*, called InferSent, uses labeled data of the Stanford Natural Language Inference and the Multi-Genre NLI datasets (*Bowman et al., 2015*; *Williams, Nangia & Bowman, 2018*) to train a Siamese bidirectional long short term memory (BiLSTM) network with max-pooling over the output. They also indicated that this sentence similarity measure consistently outperforms unsupervised methods like SkipThought. Consequently, *Cer et al. (2018)* presented a sentence embedding measure (*i.e.,* Universal Sentence Encoder) that trains a transformer network and augments unsupervised learning with training on SNLI datasets. In addition, the method presented by *Yang et al. (2018)* trains on conversations from Reddit using Siamese DAN and siamese transformer networks, which yielded good results on the STS benchmark dataset.

Web-based semantic similarity measures (*Yang et al., 2019*; *Raffel et al., 2020*), leveraging the extensive data available on the internet, have become increasingly relevant to the domain of sentence embedding measures. These approaches utilize the dynamic content of web resources, such as Wikipedia, to enhance semantic analysis and sentence similarity evaluations. By applying heuristic algorithms and semantic vector extraction techniques, web-based measures offer a complementary perspective to traditional sentence embedding methods (*Devlin et al., 2018*). For instance, emotion vector extraction, informed by psychological frameworks like those of *Ekman (1992)* and *Plutchik (1980)*, extends the capabilities of sentence embeddings to encompass emotional recognition within web content. Moreover, the analysis of semantic similarities in images and social media content through metadata and contextual understanding further aligns with the principles of sentence embedding measures. This integration of web-based semantics not only enriches the process for evaluating sentence similarity but also underscores the potential for novel applications, from emotional analysis to content similarity assessments, bridging the gap between traditional sentence embeddings and the expansive knowledge of the web (*Neumann et al., 2019*).

The family of sentence embedding measures (*Devlin et al., 2018*; *Liu et al., 2019*; *Nils & Iryna, 2019*; *Wang et al., 2023*), which have become very popular over the last few years, are based on the compression of high-dimensional vectors. Sentences are passed to the transformer network, and the target value is predicted. However, this setup requires that both compared sentences be fed into the network, which leads to a massive computational

complexity overhead. Interestingly, *Nils & Iryna (2019)* proposed a new variation of the BERT method proposed by *Devlin et al. (2018)* called SentenceBERT, which pre-trained BERT network that uses siamese and triplet network structures to derive semantically meaningful sentence embeddings. These embeddings can then be compared using cosine similarity to calculate the similarity scores between compared sentences. Compared with BERT measure, this reduces the computational complexity required in semantic similarity measurement.

Sentence embedding similarity measures are commonly evaluated on binary classification tasks such as semantic textual similarity (STS) (*Cer et al., 2018*). However, there are some issues with evaluating semantic similarity measures in this way. Firstly, performing binary classification requires a threshold to be determined, which requires a training dataset. Most researchers who have used these datasets are interested only in unsupervised learning, and usually choose a threshold of 0.5 (*Mihalcea, Corley & Strapparava, 2006*). This choice, however, is improvised, since the recent sentence embedding similarity measures do not output probabilities; moreover, some measures output a consistently higher range of values than others. Secondly, performing binary classification does not test the full discriminatory capability of a similarity measure. For example, suppose a measure achieves good performance on a classification task. In that case, it does not necessarily follow that the measure will achieve good performance when used within some other tasks, such as clustering. Since our interest in evaluating the performance of short text clustering methods is motivated by sentence embedding measures, we choose to use the SentenceBERT as a text similarity measure on a sentence clustering task.

## Clustering methods

Clustering text at the document level, however, is well-studied in the information retrieval literature, where documents are typically represented as data objects in a high-dimensional vector space in which each dimension corresponds to a unique keyword (*Salton, 1989*), leading to a rectangular representation in which rows represent documents and columns represent attributes of those documents. This type of data, known as attribute data, is responsive to clustering by an extensive range of methods (*Rakib et al., 2020*; *Jin & Bai, 2019*; *MacQueen, 1967*; *Karypis, Han & Kumar, 1999*; *Corsini, Lazzerini & Marcelloni, 2005*; *Zhao, Karypis & Fayyad, 2005*; *Luxburg, 2007*; *Frey & Dueck, 2007*; *Shams-Shemirani et al., 2023*; *Rezaei et al., 2023*). There are a variety of different approaches to sentence clustering (*Jain, Murty & Flynn, 1999*; *Ahmed et al., 2023*) that generally can be categorized into: a partitional clustering (*Wagstaff et al., 2001*; *Ng, Jordan & Weiss, 2001*; *Frey & Dueck, 2007*; *MacQueen, 1967*; *Sharma & Seal, 2021*; *Qi et al., 2021*), a hierarchical clustering (*Guo, Zhao & Liu, 2019*; *Roux, 2018*), and a fuzzy clustering (*Skabar & Abdalgader, 2013*; *Hathaway, Devenport & Bezdek, 1989*).

Partitional clustering is the most common clustering technique that based on the notation of reduce a given clustering criterion by iterative relocating sentences between classes till a best partition is achieved (*Shrestha, Jacquin & Daille, 2012*). This type of clustering distributes the sentences into $k$ partition, where each partition represents a class. A sentence belongs to a class if the semantic similarity between the vector of the sentence

and the centroid of this class is similar or related compared with the semantic similarity between the sentence vector and the centroid of the other classes. As a result, each class must contain at least one sentence, and every sentence is allocated to one and only one class. The primary drawback of partitional clustering methods, like $k$-means (*MacQueen, 1967*) and $k$-medoids (*Park & Jun, 2009*), is their sensitivity to the initial selection of centroids. If a sentence is closely aligned with the centroid of multiple classes, the algorithm can produce suboptimal results due to the overlap of sentences. To mitigate this issue, it is often necessary to run the clustering algorithm multiple times with different starting points for the centroids. To overcome these issues, *Frey & Dueck (2007)* introduced Affinity Propagation, a method that considers all sentences as potential centroids simultaneously. In the affinity propagation clustering method, each sentence is considered as a node within a network. The algorithm iteratively sends real-valued messages between these nodes, along the network's edges, until it successfully identifies a robust set of centroids and their associated clusters (*Frey & Dueck, 2007*). In this article, we used a modified version of the $k$-means clustering method due to *Abdalgader (2017)*.

Spectral clustering and its variants (*Luxburg, 2007*; *Ng, Jordan & Weiss, 2001*; *Yu & Shi, 2003*; *Sharma & Seal, 2021*; *Qi et al., 2021*) are also referred to the partitional (*i.e.,* graph-based) clustering approaches that allow more flexible distance metrics. Rather than clustering data points (*i.e.,* objects) in the original vector space, they map data points onto the space defined by the eigenvectors associated with the top eigenvectors, and then perform clustering in this transformed space, typically using a $k$-means algorithm. Formally, given a set of sentences, $S = S_1, S_2, \ldots, S_n$, the semantic similarity matrix, $M \in \mathbb{R}^{n*n}$, is computed using one of the existing sentence similarity measures. Following this, construct the affinity matrix $A \in \mathbb{R}^{n*n}$ defined by the Gaussian similarity function, and then normalize the graph Laplacian matrix. Next, calculate the eigenvector of the f Laplacian matrix, select the $k$ largest eigenvector and fill them in columns from $X = [x_1, x_2, \ldots, x_n] \in \mathbb{R}^{n*k}$. Consequently, normalize the rows of $X$ to have unit length to form the matrix Y, and finally use $k$-Means to cluster the rows of matrix Y into $k$ classes by treating them as points in $\mathbb{R}^k$. This means that the most spectral clustering algorithms need to calculate the Laplacian matrix (*i.e.,* full graph) and thus have quadratic complexities in the number of data points. However, one of the advantages of this type of clustering algorithms is that it can identify non-convex clusters, which is not possible when using $k$-means algorithms. Since most spectral clustering algorithms, however, are based on matrix decomposition techniques that require only a matrix containing the numerical scores as input, it is straightforward to apply them to sentence-level text clustering task.

Hierarchical clustering algorithms (*Guo, Zhao & Liu, 2019*; *Roux, 2018*) create a hierarchical structure of classes called a dendrogram, and this can be created in two ways: top-down (*i.e., divisive*) or bottom-up (*i.e., agglomerative*). With the top-down algorithms, all the data points are stated in the same class, and as we iterate, they are divided into smaller subsets until each data point is in an individual class or fulfills a condition of completion. In the case of bottom-up, however, the data points are successively combined according to the measurements until they are all joined into one class or meet a completion condition. In other words, the hierarchical clustering algorithms treat each observation as

an individual class. This means that they repeatedly identify the two most similar classes, and then merge them into one class. This iterative process continues until all the classes are merged together. Unlike most existing partitional clustering methods, which take explicitly the number of input classes $k$, the hierarchical clustering methods take a threshold $t$ as an input to indirectly determine the number of classes.

Fuzzy clustering algorithms enable data points to belong to all classes with different degrees of membership. The first successful fuzzy clustering algorithms are generally due to *Hathaway, Devenport & Bezdek (1989)*, *Hathaway & Bezdek (1994)*. However, fuzzy clustering of sentence-level text is complicated by the computational difficulties inherent in defining cluster centroids using conventional cluster centrality measures. Interestingly, *Skabar & Abdalgader (2013)* show how using PageRank (*Brin & Page, 1998*) as a centrality measure can be extended to multiple clusters and present a complete fuzzy relational clustering algorithm. Using PageRank in this algorithm is to determine the important of a node within a graph based on global information recursively calculated from the entire graph. It is this importance that can be used as a measure of centrality. The algorithm, therefore, allows sentences to belong to all clusters with different degrees of semantic similarity. This is important in the case of downstream applications, in which a sentence may be semantically related to more than one theme or topic in a document or set of documents.

Sentence clustering methods, while primarily employed in natural language processing and text analysis, hold promising applications across various engineering domains, notably in optimization strategies (*Taleizadeh et al., 2023*; *Gharaei et al., 2023*). The utilization of sentence clustering methodologies can be extended to engineering problems that involve optimization tasks (*Gharaei et al., 2023*). For instance, in structural engineering, clustering similar sentences extracted from design specifications can aid in categorizing and organizing information related to specific materials, structural elements, or construction techniques. By identifying clusters of similar sentences, engineers can streamline information retrieval, perform comparative analyses between different design approaches, and enhance knowledge transfer within the domain. Moreover, in optimization processes such as algorithm design or parameter tuning, sentence clustering techniques might facilitate the grouping and categorization of optimization strategies or algorithms based on their similarities, aiding in the systematic comparison, selection, and refinement of optimization techniques tailored for specific engineering problems. Therefore, integrating sentence clustering approaches into engineering optimization tasks holds the potential to streamline information management, enhance decision-making processes, and foster innovation within diverse engineering disciplines. As we concern ourselves in this section with relevant works to the text representation, sentence embedding measures, and clustering methods, Table 1 analyzes and summarizes the related techniques used in our proposed framework found in the literature.

## PRELIMINARIES

The task of sentence clustering is to take a collection of sentences as input dataset and grouping sentences into classes as output, so that sentences related to each other are

**Table 1  Comparison of related techniques to our proposed framework.**

| Category | Techniques | References | Techniques used in our proposed framework |
|---|---|---|---|
| Word similarity | Co-occurrence | *Xu et al. (2015)*, *Xu et al. (2017)*, *Patibandla & Veeranjaneyulu (2018)*, *Hadifar et al. (2019)* | No |
| | Word embedding | *Hathaway, Devenport & Bezdek (1989)*, *Salton (1989)*, *Mikolov et al. (2013b)*, *Mikolov et al. (2013a)*, *Pennington, Socher & Manning (2014)* | Yes |
| Sentence similarity | Lexical resources | *Raffel et al. (2020)*, *Islam & Inkpen (2008)*, *Li et al. (2006)* | No |
| | Sentence embedding | *Kiros et al. (2015)*, *Hill, Cho & Korhonen (2016)*, *Conneau et al. (2017)*, *Cer et al. (2018)*, *Nils & Iryna (2019)*, *Abdalgader & Skabar (2010)*, *Peters et al. (2018)*, *Bojanowski et al. (2017)*, *Pei et al. (2020)*, *Giorgi et al. (2021)*, *Wang & Kuo (2020)*, *Pragst, Minkera & Ultes (2020)*, *Perone, Silveira & Paula (2018)*, *Bowman et al. (2015)*, *Williams, Nangia & Bowman (2018)*, *Yang et al. (2018)*, *Devlin et al. (2018)* | Yes |
| | Web-based resources | *Yang et al. (2019)*, *Raffel et al. (2020)*, *Devlin et al. (2018)*, *Ekman (1992)*, *Plutchik (1980)*, *Neumann et al. (2019)* | No |
| Sentence clustering | Partitional clustering | *Mihalcea, Corley & Strapparava (2006)*, *Fellbaum (1998)*, *Wagstaff et al. (2001)*, *Neumann et al. (2019)*, *Luxburg, (2007)* | Yes |
| | Hierarchical clustering | *Ng, Jordan & Weiss (2001)*, *Frey & Dueck (2007)* | Yes |
| | Fuzzy clustering | *Skabar & Abdalgader (2013)*, *Guo, Zhao & Liu (2019)* | Yes |
| Text analysis | Machine learning | *Abdalgader (2017)*, *Luxburg (2007)*, *Hathaway & Bezdek (1994)* | No |

grouped in a unique class. Formally, consider a dataset comprising $N$ sentences denoted as $S = \{S_1, S_2, \ldots, S_N\}$, where each sentence $S_i$ is a sequence of words $S_i = \{w_{i1}, w_{i2}, \ldots, w_{il}\}$, and $l$ represents the length of sentence $S_i$. To represent sentences as feature vectors, it is straightforward to extend the idea of word embeddings to representing word sequences (*i.e.,* sentence-level text) in low-dimension vector space, as shown in Fig. 1.

There are many representation schemes used for reweighting word-level embeddings to utilize sentence-level embedding (*Wieting et al., 2016*; *Arora, Liang & Ma, 2017*; *Nils & Iryna, 2019*), and the most straightforward representation embeds a sentence $S_i$ by averaging the vectors of its words. The only parameters learned by this sentence embedding are word embedding matrix $M$ and their averaging defined as:

$$S_i = \frac{1}{N} \sum_i^N M_{w_i} \qquad (1)$$

where $M_{w_i}$ is the word embedding value for word $w_i$.

Consequently, the sentence clustering task is typically executed in two phases: initially, a semantic similarity matrix is computed based on the above described representation (*i.e.,* using SentenceBERT measure), followed by the clustering of sentences according to the scores generated from this matrix. Let $C = \{C_1, C_2, \ldots, C_k\}$ denote the cluster, where each $C_i$ represent a cluster containing sentences. In this case, the resulting clusters must meet two conditions: clusters must be as disparate as possible, and the elements that contain them as similar as possible.

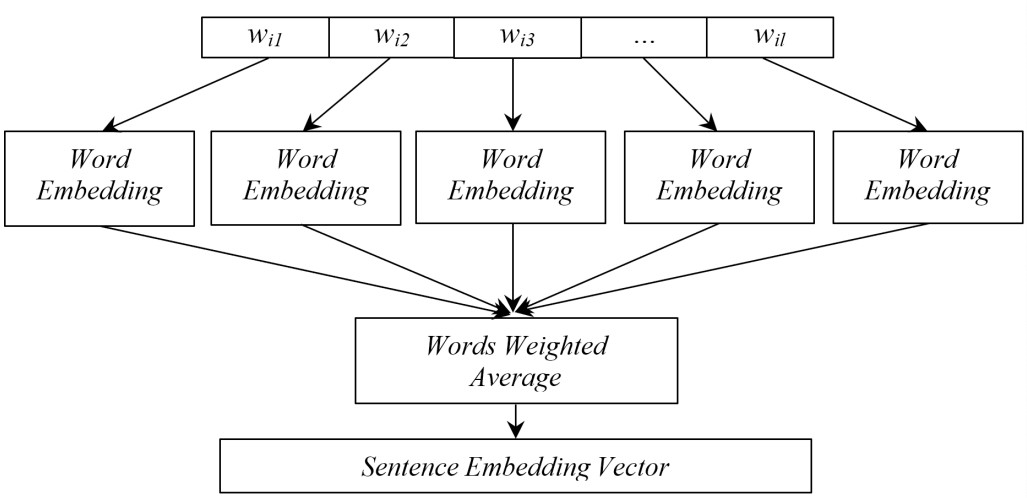

Figure 1  Simplest sentence embedding representation scheme.

# EXPERIMENTAL METHODOLOGY

The main contribution of this work is to propose an experimental and evaluation framework that allows comparison of various clustering methods among themselves using sentence embedding similarity measure with different pre-training models as shown in Fig. 2. As can be seen from that figure, we first utilize the trained embedding techniques to obtain the semantic representations, and then employ well-known supervised algorithms to perform sentence clustering. This section, therefore, describes the methodology steps, experimental parameter settings, used clustering benchmark datasets and clustering evaluation criteria.

**Pre-training model**: In the first step of our methodology, a suitable pre-training model was chosen to generate high-quality sentence embeddings. The selection of the pre-training model depends on the nature of the task and the target dataset. There are various pre-training models available for capturing semantic and contextual information, and each of these models has its strengths. The choice is made based on their ability to capture such information and semantic meaning effectively. The selected pre-training model should have been trained on a large and diverse corpus to ensure the embeddings' robustness and ability to represent sentences accurately. More details on the used models for pre-training process in our experiments are presented in 'Parameter settings'.

**Clustering dataset**: With the pre-training model chosen, the next step involves loading the target clustering dataset. This dataset consists of the sentences that must be clustered based on their semantic similarities. The dataset should represent the task at hand and contain sufficient variation to effectively evaluate the clustering performance. Depending on the application, the dataset could comprise product reviews, news articles, customer feedback, or any other collection of sentences with a common theme. In our experiments, we used the clustering benchmark datasets described in 'Clustering benchmark datasets'.

**Calculate the sentence embedding vectors**: Once the dataset is loaded, each sentence in the dataset is processed through the selected pre-training model to obtain its corresponding

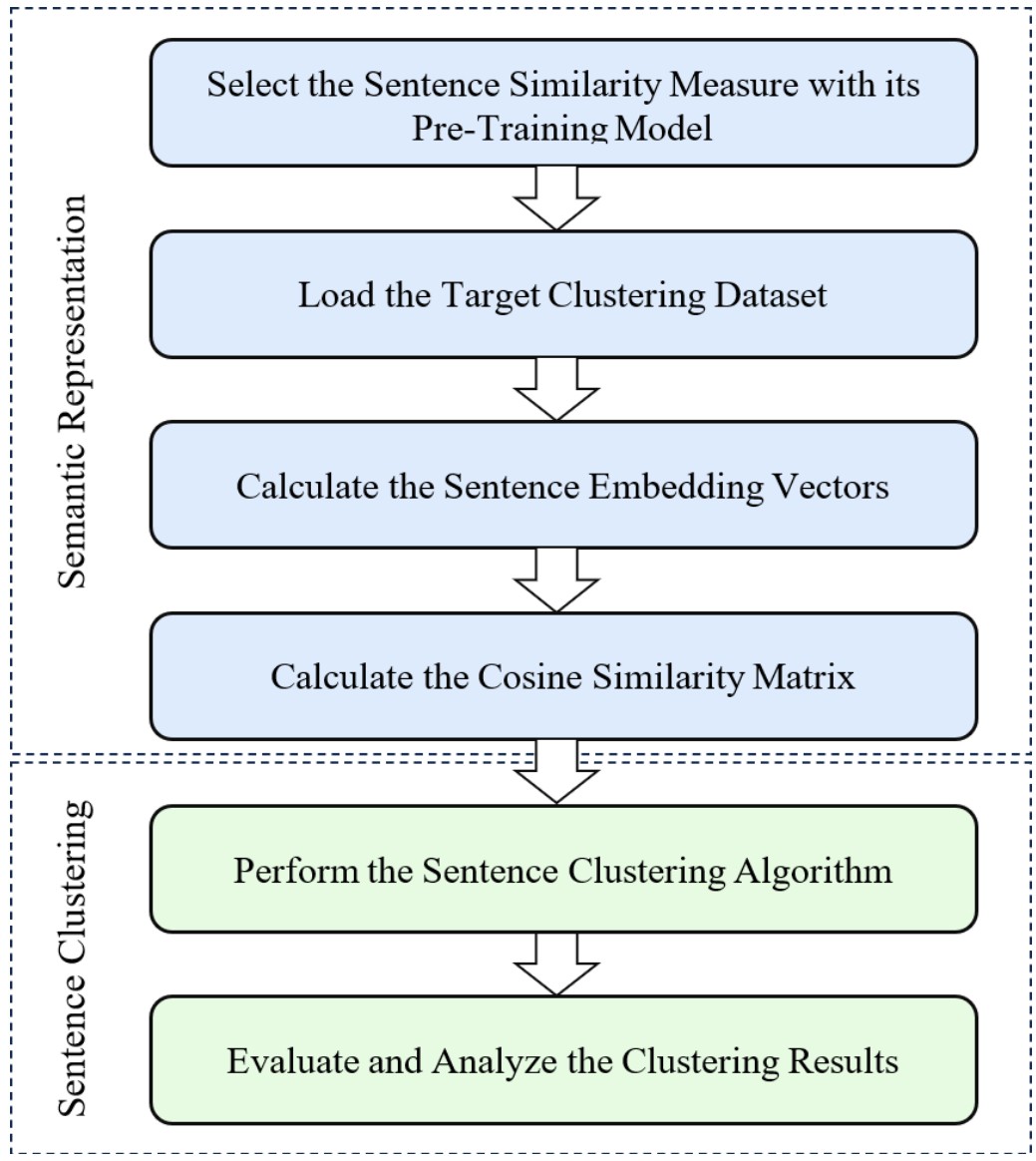

**Figure 2** Performance evaluation framework for sentence clustering method using embedding-based similarity measure.

embedding vector, as shown in Fig. 1. The embedding vector represents the sentence's semantic meaning in a dense numerical form, capturing the relationships between words and their contextual information. This step involves utilizing the pre-trained model's capabilities to generate meaningful and high-dimensional embeddings for each sentence in the dataset. The output of this step is a set of sentence embedding vectors, each representing a unique sentence in the dataset. For clarity, we apply a very effective method for embedding the target sentences, called smooth inverse frequency (SIF) embeddings. This method uses a weighted average of pre-training embedding of each word and applies a dimensionality reduction to obtain the sentence embedding vector, as shown in Fig. 3. The contribution

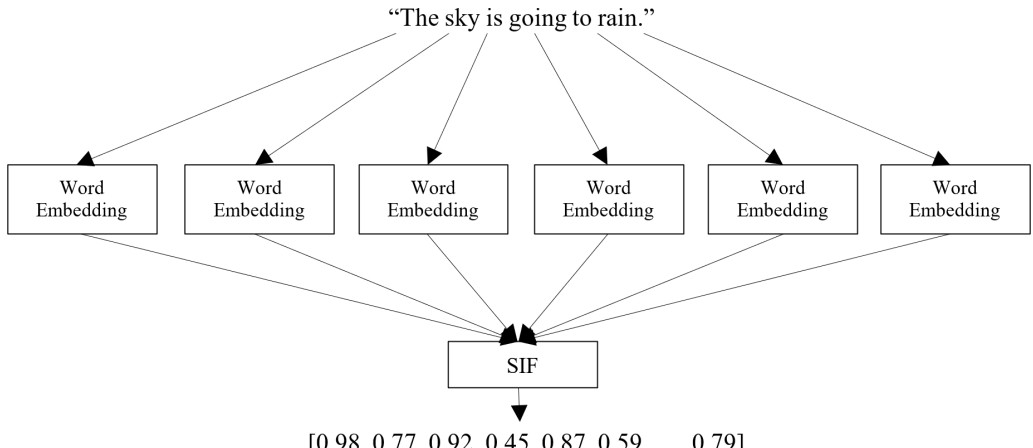

**Figure 3** Calculating a sentence embedding vector.

of each word is computed as $\frac{a}{a+p(w)}$ with $a$ being a hyperparameter and $p(w)$ being the empirical word frequency in the target dataset. SIF embeddings are then produced by computing the first principal component of all the resulting vectors and removing it from the weighted embeddings.

**Calculate the cosine similarity matrix**: Having obtained the sentence embedding vectors, the next step involves calculating the cosine similarity between all pairs of sentences embedding vectors. The cosine similarity measures the cosine of the angle between two vectors and provides a value between $-1$ and 1, indicating the similarity between the vectors in the high-dimensional space. By computing the cosine similarity for all sentence pairs, a symmetric similarity $n \times n$ matrix is created, where each element $(i, j)$ represents the similarity between sentence vector $i$ and sentence vector $j$. This similarity matrix is the basis for clustering the sentences in the subsequent step.

**Perform the sentence clustering**: With the similarity matrix in hand, a clustering algorithm is chosen to group similar sentences. For simplicity, given a set of input sentences denoted as $S = \{S_1, S_2, S_3, \ldots, S_n\}$, the goal is to partition them into clusters $C = \{C_1, C_2, C_3, \ldots, C_k\}$, where each cluster $C_k$ contains semantically similar sentences. Various clustering algorithms, such as $k$-means, agglomerative hierarchical clustering, and others, can be employed. The choice of the clustering algorithm depends on the specific characteristics of the dataset and the desired outcome of the clustering task. For instance, $k$-means aims to partition the data into $k$ clusters by finding centroids, while agglomerative hierarchical clustering merges similar clusters iteratively. The selected algorithm is applied to the similarity matrix to form coherent clusters of similar sentences. See 'Parameter settings' for more details on specific clustering algorithms used in our experiments.

**Evaluate and analyze the clustering results**: The final step involves evaluating the quality of the obtained clusters and analyzing the clustering results. Since the ground truth labels are available (*e.g.*, for a supervised clustering task) in the used benchmark datasets, evaluation metrics like the Adjusted Rand Index (ARI) (*Rand, 1971*), Fowlkes-Mallows

Index (FMI) (*Fowlkes & Mallows, 1983*), Normalized Mutual Information (NMI) (*Danon, Díaz-Guilera & Duch, 2005*) can be used to measure the agreement between the obtained clusters and the ground truth labels. Additionally, a Silhouette Score (SIL) (*Rousseeuw, 1987*) can be conducted by inspecting the clustering results to understand whether the clusters make sense in the context of the task and to gain insights into the effectiveness of the clustering approach. More details on the clustering evaluation criteria are discussed in 'Clustering evaluation criteria'.

## Parameter settings

We pre-processed the target textual datasets by removing noises and tokenizing the texts into sentences. Then, we transformed each sentence into the embedding vector representation using a pre-training sentence embedding model. The pre-training model used in our experiments for measuring the semantic similarity between compared sentences was the sentence-transformer model called SentenceBERT (*Nils & Iryna, 2019*). We also provided results of using a traditional sentence similarity measure (*i.e.,* without embedding techniques) (*Abdalgader & Skabar, 2010*) for the comparative purpose. Since we evaluate our clustering framework on textual datasets come from different sources, the pre-trained BERT model has been trained based on two different models known as "*bert-base-nli-mean-tokens*" and "*paraphrase-distilroberta-base-v1*". This is to avoid the impact of BERT model selection on the performance of text clustering and to explore their effects on each type of this dataset. In all experiments, however, we set the size of sentence vector dimension as $d = 200$ as we tuned different dimensions for sentence vectors. When the size is small (less than 100), performance drops significantly. When the size is larger (more than 300), the curve flattens out. To make our model more efficient, we fixed it as 200.

Three different clustering methods have been applied to cluster the target texts as described in the step of performing the sentence clustering in the above section. The used clustering methods are $k$-means (*Abdalgader, 2017*) for partitional clustering, agglomerative (*Roux, 2018*) for hierarchical clustering and fuzzy-relational (*Skabar & Abdalgader, 2013*) for fuzzy clustering. The number of clusters is the same number of labels in the dataset. The clustering performance is evaluated with four metrics described in 'Clustering evaluation criteria' To show the statistical significance, the performance of each experiment is the average of 50 trials.

## Clustering benchmark datasets

Three popular datasets for text clustering: MR (*Pang & Lee, 2005*), AG News (*Zhang & LeCun, 2016*) and SearchSnippets (*Phan, Nguyen & Horiguchi, 2008*) are used to evaluate our proposed sentence clustering framework. These datasets have been selected in our experiments because they are commonly used in cluster tasks, rich in context, and substantial in size, enabling comprehensive analysis and evaluation of our evaluation framework. The statistics of these datasets are presented in Table 2, and the detailed descriptions are the following:

MR (*Pang & Lee, 2005*): The MR is a movie review dataset for binary sentiment classification, containing 1,777 positive and 1,777 negative reviews.

**Table 2  Statistics for the clustering benchmark datasets.**

| Dataset | Categories | Samples | Average length | Vocabulary |
|---|---|---|---|---|
| MR | 2 | 3,554 | 21.02 | 18,196 |
| AG News | 4 | 7,600 | 6.76 | 16,406 |
| SearchSnippets | 8 | 2,280 | 17.9 | 30,610 |

AG News (*Zhang & LeCun, 2016*): The AG's news topic classification dataset contains around 127,600 English news articles, labeled with four categories. We use the test dataset for experiments, which includes 7,600 news titles.

SearchSnippets (*Phan, Nguyen & Horiguchi, 2008*): A collection consisting of web search snippets categorized into eight domains. The eight domains are Business, Computers, Culture-Arts, Education-Science, Engineering, Health, Politics-Society, and Sports.

## Clustering evaluation criteria

We used four different metrics to evaluate the clustering performance of our experimental methodology. The first metric SIL, falls under the category of internal or supervised criteria (when true labels are unknown) for clustering evaluation, which measures clustering quality. The other three metrics, known as the ARI, FMI, and NMI fall under the category of external or *unsupervised* criteria (when true labels are available) for clustering evaluation, which measure clustering membership. The following are detailed descriptions of these used criteria.

**Silhouette score**: While not strictly an external evaluation metric, the silhouette score (*Rousseeuw, 1987*) measures the quality of individual clusters. It considers the distance between the samples within the same cluster and the distance between the samples in different clusters and is defined as:

$$Silhouette_{Score} = \frac{1}{N} \sum_i \frac{b_i - a_i}{max\{a_i, b_i\}}, \tag{2}$$

where $N$ is the number of data points (*i.e.*, sentences), $a_i$ is the average distance between $i$ and all other sentences in the same cluster, and $b_i$ is the smallest average distance of $i$ to any other cluster, minimizing the distance to other clusters.

**Adjusted Rand Index (ARI)**: ARI quantifies the similarity between the true and predicted clusters while correcting for chance (*Rand, 1971*). It measures the proportion of pairs of texts that are either in the same cluster in both the true and predicted clusters or in different clusters in both. ARI values range from -1 (random clustering) to 1 (perfect clustering), and we compute it using the following equation:

$$ARI = \frac{RI - Expected\_RI}{max(RI\_max - Expecteed\_RI, 0)}, \tag{3}$$

where $RI$ is the Rand Index, *Expected RI* is the expected Rand Index under a random clustering, and *RI_max* is the maximum possible Rand Index.

**Fowlkes-Mallows Index (FMI)**: FMI computes the geometric mean of precision and recall between the true and predicted clusters (*Fowlkes & Mallows, 1983*). It considers both

the number of true positives and false positives. FMI values range from 0 (no similarity) to 1 (perfect similarity), and can be calculated using the following equation:

$$FMI = \frac{TP}{\sqrt{TP+FP} * \sqrt{TP+FN}},$$ (4)

where $TP$ is the number of true positives, $FP$ is the number of false positives, and $FN$ is the number of false negatives.

**Normalized Mutual Information** (**NMI**): NMI measures the mutual information between the predicted clusters and the true categories of the texts, normalized to account for chance (*Danon, Díaz-Guilera & Duch, 2005*). It considers both the information gained and the entropy of the clusters. NMI values range from 0 (no mutual information) to 1 (perfect clustering), and also can be computed using the following equation:

$$NMI = \frac{2 * MI}{H(C) + H(T)},$$ (5)

where $MI$ is the mutual information between predicted clusters $C$ and true categories $T$, and $H$ is the entropy.

# EMPIRICAL RESULTS

In order to evaluate the effectiveness of the proposed clustering evaluation framework, we apply it to two different scenarios of experiments. Firstly, we evaluate the framework (*i.e.,* three selected clustering methods) as a stand-alone (*i.e., in vitro*) on several standard short-text clustering datasets. We then apply the best performed method on end-to-end tasks (*i.e., in vivo*), involving document summarization task.

## Results on benchmark datasets

In this section, we conduct the experiment to evaluate the effectiveness of the $k$-means (*Abdalgader, 2017*), agglomerative (*Roux, 2018*) and fuzzy-relational (*Skabar & Abdalgader, 2013*) methods as a stand-alone using three benchmark datasets: MR (*Pang & Lee, 2005*), AG News (*Zhang & LeCun, 2016*) and SearchSnippets (*Phan, Nguyen & Horiguchi, 2008*). We first discuss the obtained results and compare the performance of the used clustering methods with other existing approaches reported in *Yin et al. (2021)*. Then, we apply the best performed clustering method to a text summarization task as an end-to-end evaluation in 'Results on MR dataset'.

### Results on MR dataset

Tables 3 and 4 present the performance results of three text clustering methods: $k$-means, agglomerative, and fuzzy-relational, applied to MR dataset. These experiments utilized two distinct pre-trained BERT models: "*bert-base-nli-mean-tokens*" (Table 3) and "*paraphrase-distilroberta-base-v1*" (Table 4). The first section of each table showcases the performance of the SentenceBERT measure (*Nils & Iryna, 2019*), which employs embedding techniques for sentence semantic similarity measurement. The second section reveals the performance results without embedding techniques, where sentence similarity relies solely on the method developed by *Abdalgader & Skabar (2010)*.

**Table 3  Performance results on the MR dataset using "bert-base-nli-mean-tokens" Pre-trained BERT model.**

| Clustering methods | SIL | NMI | ARI | FMI |
|---|---|---|---|---|
| Sentence similarity measure with embedding techniques | | | | |
| Partitional clustering ($k$-means) | 0.224 | 0.254 | 0.516 | 0.528 |
| Hierarchical clustering (Agglomerative) | 0.055 | 0.139 | 0.583 | 0.598 |
| Fuzzy clustering (Fuzzy-Relational) | 0.211 | 0.041 | 0.527 | 0.530 |
| Sentence similarity measure without embedding techniques | | | | |
| Partitional clustering ($k$-means) | 0.105 | 0.110 | 0.344 | 0.297 |
| Hierarchical clustering (Agglomerative) | 0.001 | 0.021 | 0.401 | 0.349 |
| Fuzzy clustering (Fuzzy-Relational) | 0.099 | 0.002 | 0.301 | 0.289 |

**Table 4  Performance results on the MR dataset using "paraphrase-distilroberta-base-v1" Pretrained BERT model.**

| Clustering methods | SIL | NMI | ARI | FMI |
|---|---|---|---|---|
| Sentence similarity measure *with* embedding techniques | | | | |
| Partitional clustering ($k$-means) | 0.164 | 0.292 | 0.499 | 0.501 |
| Hierarchical clustering (Agglomerative) | 0.000 | 0.000 | 0.499 | 0.563 |
| Fuzzy clustering (Fuzzy-Relational) | 0.155 | 0.000 | 0.500 | 0.500 |
| Sentence similarity measure without embedding techniques | | | | |
| Partitional clustering ($k$-means) | 0.091 | 0.123 | 0.279 | 0.313 |
| Hierarchical clustering (Agglomerative) | −0.133 | −0.099 | 0.295 | 0.344 |
| Fuzzy clustering (Fuzzy-Relational) | 0.021 | −0.087 | 0.329 | 0.282 |

When SentenceBERT is used, the "*bert-base-nli-mean-tokens*" pre-trained model consistently demonstrates superior performance across all clustering methods, as indicated by higher values for overall SIL, NMI, ARI, and FMI. In contrast, omitting embedding techniques results in a substantial decrease in SIL, NMI, ARI, and FMI values across all clustering methods. This highlights the significance of incorporating embedding techniques for measuring semantic similarity in text clustering, leading to improved performance. Additionally, it is evident that the "*bert-base-nli-mean-tokens*" pre-trained BERT model outperforms the "*paraphrase-distilroberta-base-v1*" pre-trained model on this dataset. In the context of our clustering analysis, however, using two distinct pre-trained BERT models, specifically "*bert-base-nli-mean-tokens*" and "*paraphrase-distilroberta-base-v1*". We have visualized the performance results for both partitional ($k$-means) and hierarchical (agglomerative) clustering methods in Figs. 4, 5 and 6, as they pertain to hard clustering. It is worth noting that we have chosen not to include the results of fuzzy clustering (fuzzy-relational) in our visualizations, a decision driven by the considerations of maintaining consistency and conserving space limitation of this article. This is applied for all the experimental visualizations of the remaining results, where one colour denotes a single cluster.

As can be seen from Fig. 4A, by using the "*bert-base-nli-mean-tokens*" BERT model, the partitional clustering ($k$-means) achieved a Silhouette score of 0.224, indicating a moderate level of cluster cohesion and separation as also shown in Figs. 5A and 6A. The visualized

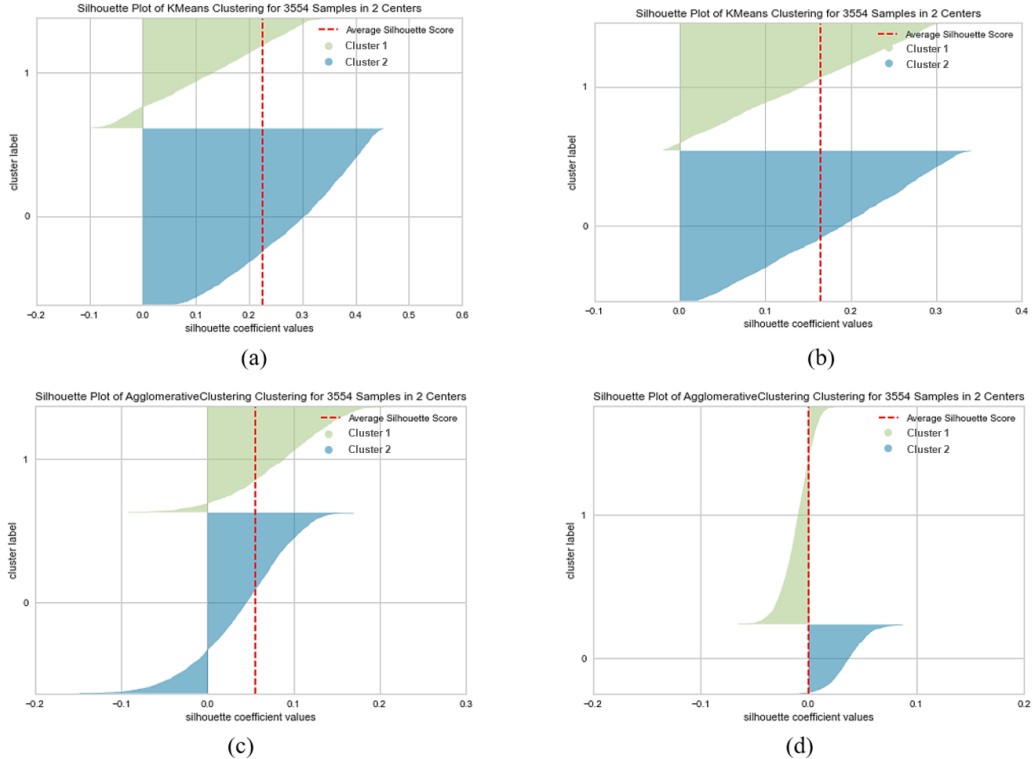

**Figure 4** *K*-means and agglomerative clustering results of the used SentenceBERT measure on the MR dataset, where one colour represent a cluster. Graphs (A) and (B) show the silhouette performance of *k*-means method using pre-trained BERT models "*bert-base-nli-mean-tokens*" and "*paraphrase-distilroberta-base-v1*" respectively. Graphs (C) and (D) show the Silhouette performance of the agglomerative method using pre-trained BERT models "*bert-base-nli-mean-tokens*" and "*paraphrase-distilroberta-base-v1*" respectively.

figures also show that normalized mutual information (NMI) was 0.254, suggesting a moderate level of mutual information between clusters and true labels. The ARI was 0.516, indicating substantial agreement between clusters and true labels, while the FMI stood at 0.528, suggesting a relatively strong clustering performance. In the case of hierarchical clustering (agglomerative), however, using the same BERT model, the Silhouette score in Fig. 4C was 0.055, indicating a lower level of cluster cohesion and separation as can also be seen from Figs. 5C and 6C. In this case, the NMI was 0.139, suggesting a relatively low level of mutual information between clusters and true labels. However, the ARI was 0.583, indicating substantial agreement between clusters and true labels, and the FMI was 0.598, suggesting a strong clustering performance.

When employing the second pre-training model "*paraphrase-distilroberta-base-v1*" for partitional clustering (*k*-means), the Silhouette score in Fig. 4B was 0.164, indicating a moderate level of cluster cohesion and separation, as shown in Figs. 5B and 6B. The NMI was 0.292, suggesting a moderate level of mutual information between clusters and true labels. The ARI was 0.499, indicating substantial agreement between clusters and true labels, and the FMI was 0.501, suggesting a relatively strong clustering performance.

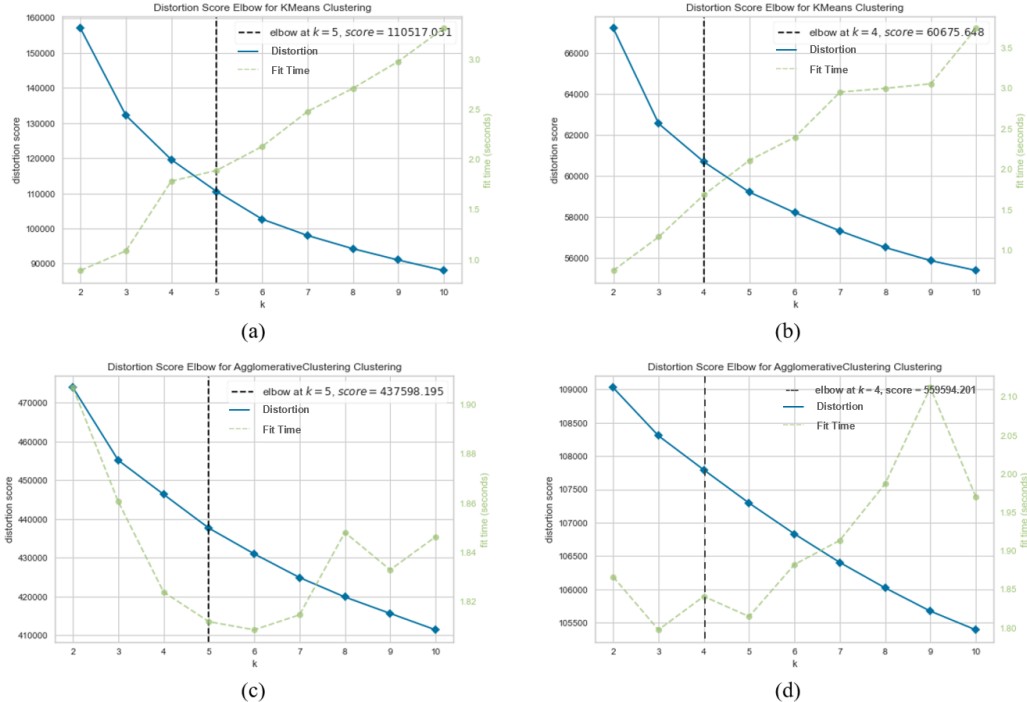

**Figure 5** *K*-means and agglomerative clustering results of the used SentenceBERT measure on MR dataset. Graphs (A) and (B) show the distortion Score performance of *k*-means method using pre-trained BERT models "*bert-base-nli-mean-tokens*" and "*paraphrase-distilroberta-base-v1*", respectively. Graphs (C) and (D) show the distortion Score performance of agglomerative method using pre-trained BERT models "*bert-base-nli-mean-tokens*" and "*paraphrase-distilroberta-base-v1*" respectively.

For hierarchical clustering (agglomerative), however, using the same pre-training model, the Silhouette score in Fig. 4D was 0.000, indicating no evident cluster cohesion and separation as shown in Fig. 5D and 6D. The NMI and ARI were 0.000, suggesting little mutual information and agreement with true labels. However, the Fowlkes-Mallows Index was 0.563, indicating a moderate clustering performance. These results finally provided a comprehensive evaluation of the clustering performance using different BERT models and clustering methods on the MR dataset, highlighting their strengths and weaknesses in various metrics.

### Results on AG news dataset

In Tables 5 and 6, we report the performance of *k*-means, agglomerative, and fuzzy-relational clustering methods on the AG News dataset, utilizing again two distinct pre-trained BERT models, specifically "*bert-base-nli-mean-tokens*" and "*paraphrase-distilroberta-base-v1*". For the first model, the results are as follows: when considering the SentenceBERT measure, the *k*-means clustering method has achieved a commendable Silhouette Score of 0.160, NMI of 0.101, ARI of 0.652, and FMI of 0.327. In parallel, the agglomerative clustering method has exhibited a Silhouette Score of 0.041, NMI of 0.319, ARI of 0.700, and FMI of 0.457. Concurrently, the fuzzy-relational clustering method has rendered a Silhouette Score of 0.058, NMI of 0.065, ARI of 0.601, and FMI of 0.343.

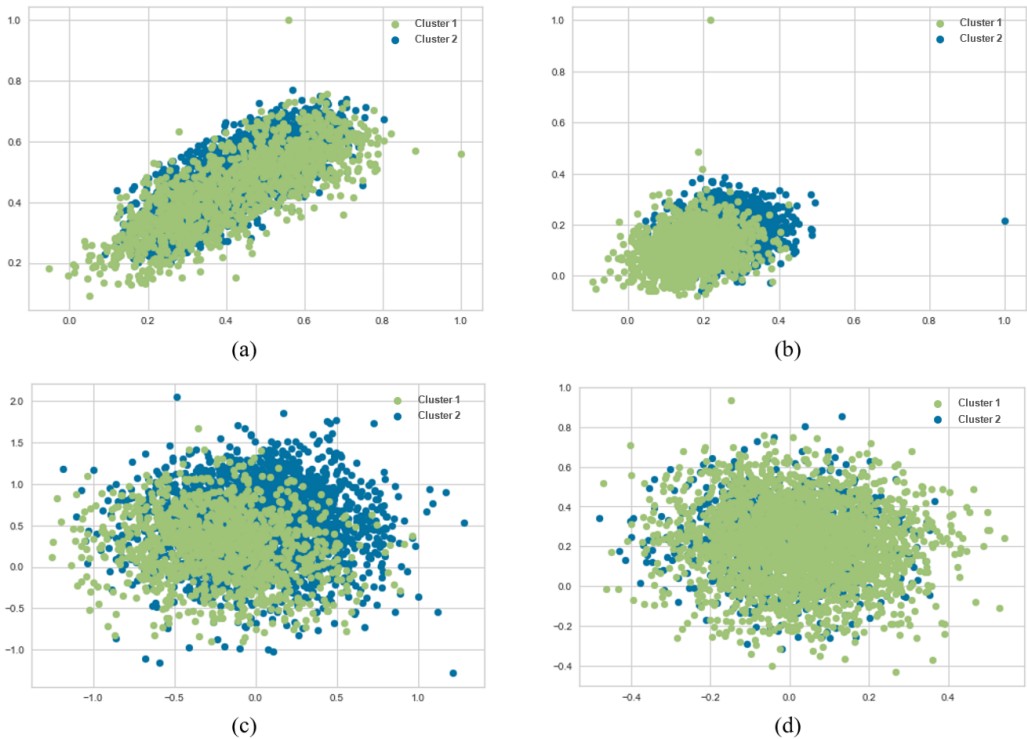

**Figure 6** *K-means and agglomerative clustering results of the used SentenceBERT measure on MR dataset, where one colour represent a cluster.* Graphs (A) and (B) show the distribution performance of *k*-means method using pre-trained BERT models "*bert-base-nli-mean-tokens*" and "*paraphrase-distilroberta-base-v1*" respectively. Graphs (C) and (D) show the distribution performance of agglomerative method using pre-trained BERT models "*bert-base-nli-mean-tokens*" and "*paraphrase-distilroberta-base-v1*", respectively.

Subsequently, when embedding techniques are not incorporated, the *k*-means method has achieved a Silhouette Score of 0.070, NMI of 0.002, ARI of 0.477, and FMI of 0.198. For the agglomerative method, the Silhouette Score manifested at 0.010, NMI at 0.129, ARI at 0.387, and FMI at 0.302. Meanwhile, the fuzzy-relational method obtained a Silhouette Score of 0.018, NMI of 0.011, ARI of 0.291, and FMI of 0.202.

Moving to the second model, the clustering performances transpire as follows: in the case of using the SentenceBERT measure, the *k*-means method modality has secured a Silhouette Score of 0.065, NMI of 0.272, ARI of 0.707, and FMI of 0.427. Concurrently, the agglomerative method has achieved a Silhouette Score of 0.007, NMI of 0.393, ARI of 0.740, and FMI of 0.541. At the same time, the fuzzy-relational method has obtained a Silhouette Score of 0.038, NMI of 0.148, ARI of 0.616, and FMI of 0.362. In the absence of embedding techniques, however, the *k*-means method modality has realized a Silhouette Score of 0.009, NMI of 0.122, ARI of 0.301, and FMI of 0.245. The agglomerative method has shown a Silhouette Score of $-0.098$, NMI of 0.193, ARI of 0.354, and FMI of 0.233. Finally, the fuzzy-relational method has indicated a Silhouette Score of $-0.050$, NMI of 0.092, ARI of 0.323, and FMI of 0.175. These empirically derived results show that

**Table 5  Performance results on the AG News dataset using "bert-base-nli-mean-tokens" Pretrained BERT model.**

| Clustering methods | SIL | NMI | ARI | FMI |
|---|---|---|---|---|
| Sentence similarity measure with embedding techniques | | | | |
| Partitional clustering (*k*-means) | 0.160 | 0.101 | 0.652 | 0.327 |
| Hierarchical clustering (Agglomerative) | 0.041 | 0.319 | 0.700 | 0.457 |
| Fuzzy clustering (Fuzzy-Relational) | 0.058 | 0.065 | 0.601 | 0.343 |
| Sentence similarity measure without embedding techniques | | | | |
| Partitional clustering (*k*-means) | 0.070 | 0.002 | 0.477 | 0.198 |
| Hierarchical clustering (Agglomerative) | 0.010 | 0.129 | 0.387 | 0.302 |
| Fuzzy clustering (Fuzzy-Relational) | 0.018 | 0.011 | 0.291 | 0.202 |

**Table 6  Performance results on the AG News dataset using "paraphrase-distilroberta-base-v1" Pre-trained BERT model.**

| Clustering methods | SIL | NMI | ARI | FMI |
|---|---|---|---|---|
| Sentence similarity measure with embedding techniques | | | | |
| Partitional clustering (*k*-means) | 0.065 | 0.272 | 0.707 | 0.427 |
| Hierarchical clustering (Agglomerative) | 0.007 | 0.393 | 0.740 | 0.541 |
| Fuzzy clustering (Fuzzy-Relational) | 0.038 | 0.148 | 0.616 | 0.362 |
| Sentence similarity measure without embedding techniques | | | | |
| Partitional clustering (*k*-means) | 0.009 | 0.122 | 0.301 | 0.245 |
| Hierarchical clustering (Agglomerative) | −0.098 | 0.193 | 0.354 | 0.233 |
| Fuzzy clustering (Fuzzy-Relational) | −0.050 | 0.092 | 0.323 | 0.175 |

the "*paraphrase-distilroberta-base-v1*" model outperforms the "*bert-base-nli-mean-tokens*" model on this dataset, except in Silhouette Scores, where "*bert-base-nli-mean-tokens*" model performs better. Such detailed analyses aid in ascertaining the models' appropriateness and effectiveness in the context of the used clustering methods.

Figures 7, 8 and 9 visualize the clustering performance results obtained using two different pre-trained BERT models, "*bert-base-nli-mean-tokens*" and "*paraphrase-distilroberta-base-v1*" on the AG News dataset using only *k*-means and agglomerative methods. In Figs. 7A, 8A and 9A, we present the *k*-means clustering performance results when utilizing the "*bert-base-nli-mean-tokens*" model. As can be seen from these figures, the SIL score is 0.160, indicating a moderate level of cluster separation. The NMI score of 0.101 suggests a limited degree of mutual information between the true labels and cluster assignments. The ARI of 0.652 demonstrates moderate agreement between true labels and cluster assignments, while the FMI of 0.327 reflects a moderate balance between precision and recall. In contrast, the agglomerative method with the same pre-trained model in Figs. 7C, 8C and 9C yields a lower SIL score of 0.041, indicating reduced cluster separation compared to *k*-means. However, the NMI (0.319) suggests a higher degree of mutual information between true labels and cluster assignments. The ARI (0.700) signifies a more significant agreement between true labels and cluster assignments compared to *k*-means, and the FMI (0.457) reflects a moderate balance between precision and recall.

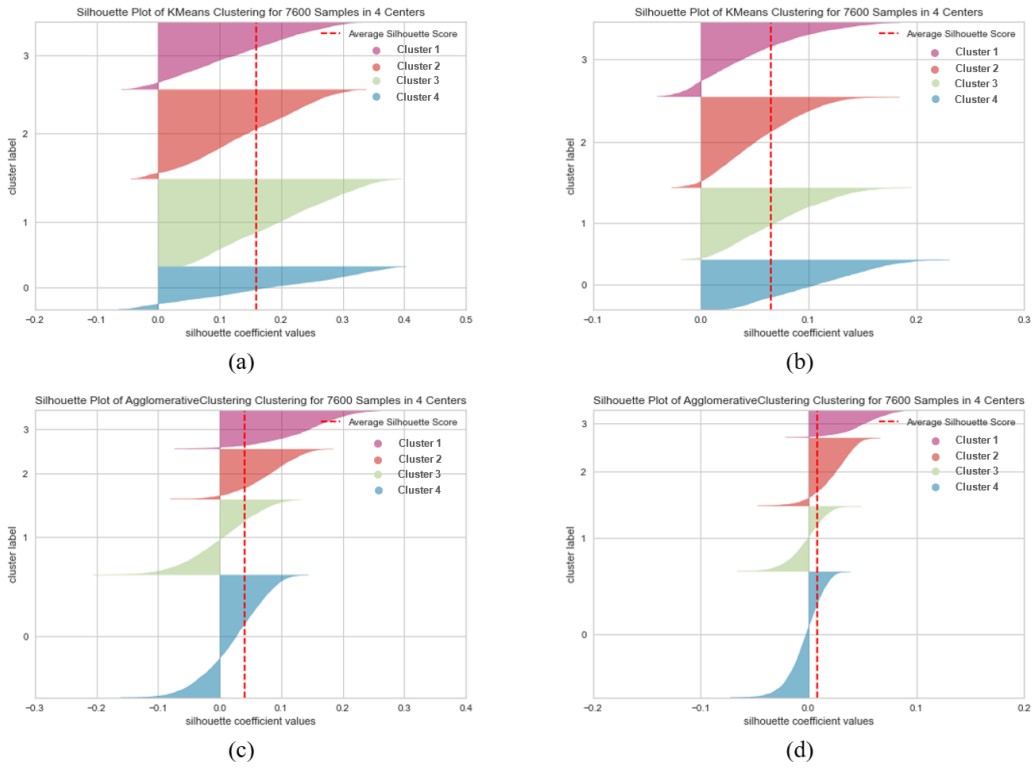

**Figure 7** *K*-means and agglomerative clustering results of the used SentenceBERT measure on AG News dataset, where one colour represents a cluster. Graphs (A) and (B) show the silhouette performance of *k*-means method using pre-trained BERT models "*bert-base-nli-mean-tokens*" and "*paraphrase-distilroberta-base-v1*" respectively. Graphs (C) and (D) show the Silhouette performance of the agglomerative method using pre-trained BERT models "*bert-base-nli-mean-tokens*" and "*paraphrase-distilroberta-base-v1*", respectively.

In Figs. 7B, 8B and 9B, we present the *k*-means clustering performance results when utilizing the "*paraphrase-distilroberta-base-v1*" model. As can be seen from these figures, the SIL score is 0.065, indicating a moderate level of cluster separation. The NMI score of 0.272 suggests a moderate degree of mutual information between true labels and cluster assignments. The ARI of 0.707 demonstrates moderate agreement between true labels and cluster assignments, while the FMI of 0.427 reflects a moderate balance between precision and recall. In the case of agglomerative with the same pre-trained model (Figs. 7B, 8B and 9B), a notably lower SIL score of 0.007 is observed, signifying minimal cluster separation. However, the NMI (0.393) suggests a higher degree of mutual information between true labels and cluster assignments compared to *k*-means. The ARI (0.740) indicates a superior agreement between true labels and cluster assignments, while the FMI (0.541) reflects a moderate balance between precision and recall. The choice of pre-trained model and clustering method, therefore, should be made judiciously, considering the specific objectives of the clustering task and the trade-offs between separation and distribution metrics. These findings contribute valuable insights for researchers and practitioners in natural language processing and clustering applications.

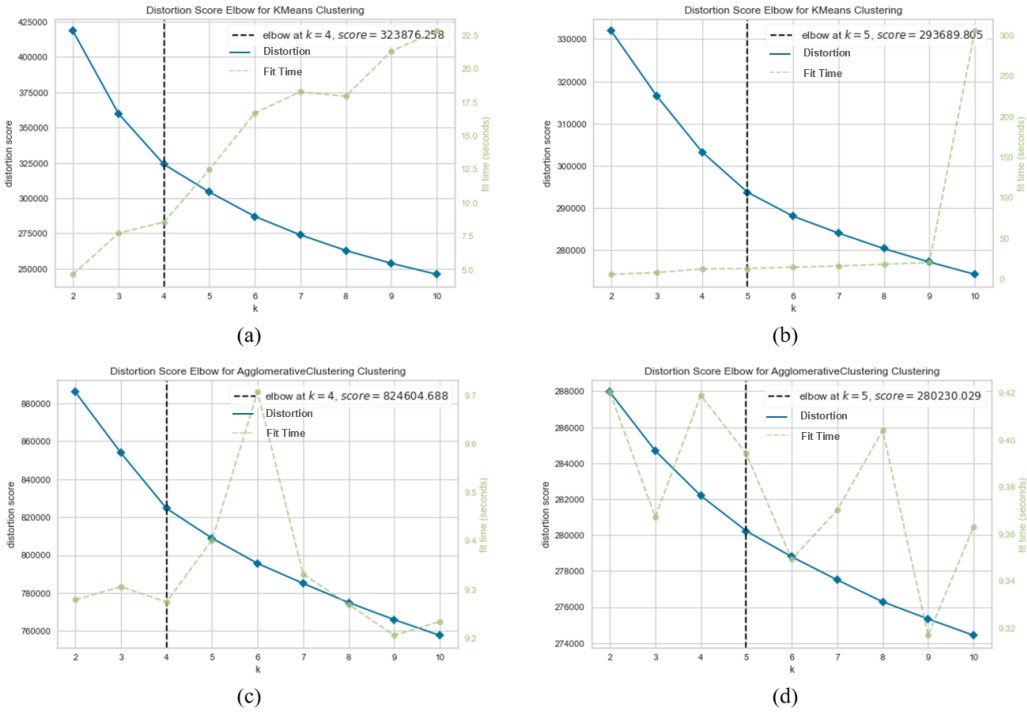

**Figure 8  K-means and agglomerative clustering results of the used SentenceBERT measure on AG News dataset.** Graphs (A) and (B) show the distortion Score performance of k-means method using pre-trained BERT models "*bert-base-nli-mean-tokens*" and "*paraphrase-distilroberta-base-v1*" respectively. Graphs (C) and (D) show the distortion Score performance of the agglomerative method using pre-trained BERT models "*bert-base-nli-mean-tokens*" and "*paraphrase-distilroberta-base-v1*", respectively.

### Results on SearchSnippets dataset

Tables 7 and 8 present the clustering performance results on the SearchSnippets dataset using two different pre-trained BERT models, "*bert-base-nli-mean-tokens*" and "*paraphrase-distilroberta-base-v1*" respectively. For the "*bert-base-nli-mean-tokens*" model, it is evident that the agglomerative method outperforms other methods in terms of NMI (0.807) and ARI (0.951), indicating strong cluster cohesion and alignment with ground truth labels. The k-means method also performs reasonably well with NMI and ARI values of 0.588 and 0.896, respectively, suggesting its effectiveness in forming well-separated clusters. However, the fuzzy-relational method struggles, showing a negative Silhouette Score and relatively lower NMI, ARI, and FMI values, indicating challenges in defining clear cluster boundaries. Furthermore, the absence of embedding techniques adversely affects clustering performance. In contrast, when utilizing the "*paraphrase-distilroberta-base-v1*" model, the agglomerative method excels, achieving the highest NMI (0.861) and ARI (0.968) among all methods, highlighting its superior ability to form coherent clusters. The k-means method also shows improvement in SIL (0.107) and NMI (0.682) compared to the previous model, indicating enhanced cluster quality. Nevertheless, the fuzzy-relational method still lags behind in performance metrics. These results underscore the critical role of the choice

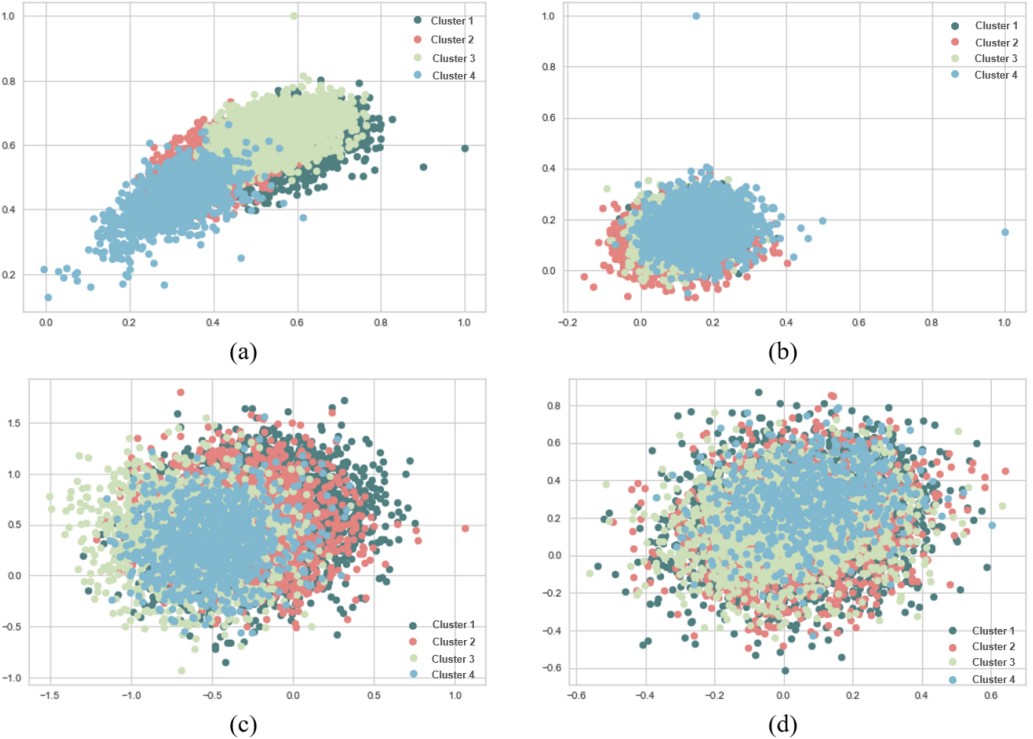

**Figure 9** *K*-means and agglomerative clustering results of the used SentenceBERT measure on AG News dataset, where one colour represent a cluster. Graphs (A) and (B) show the distribution performance of *k*-means method using pre-trained BERT models "*bert-base-nli-mean-tokens*" and "*paraphrase-distilroberta-base-v1*" respectively. Graphs (C) and (D) show the distribution performance of agglomerative method using pre-trained BERT models "*bert-base-nli-mean-tokens*" and "*paraphrase-distilroberta-base-v1*", respectively.

**Table 7** Performance results on the SearchSnippets dataset using "bert-base-nli-mean-tokens" pre-trained BERT model.

| Clustering methods | SIL | NMI | ARI | FMI |
|---|---|---|---|---|
| *Sentence similarity measure with embedding techniques* | | | | |
| Partitional clustering (*k*-means) | 0.153 | 0.588 | 0.896 | 0.599 |
| Hierarchical clustering (Agglomerative) | 0.081 | 0.807 | 0.951 | 0.815 |
| Fuzzy clustering (Fuzzy-Relational) | −0.067 | 0.216 | 0.592 | 0.358 |
| *Sentence similarity measure without embedding techniques* | | | | |
| Partitional clustering (*k*-means) | 0.076 | 0.290 | 0.439 | 0.302 |
| Hierarchical clustering (Agglomerative) | −0.090 | 0.450 | 0.422 | 0.390 |
| Fuzzy clustering (Fuzzy-Relational) | −0.020 | 0.132 | 0.294 | 0.225 |

of a pre-trained BERT model and the utilization of embedding techniques in achieving high-quality clustering results. The suboptimal performance of the fuzzy-relational method in both scenarios may be due to its sensitivity to noise and its inherent fuzziness, making it less suitable for tasks with well-defined clusters like text clustering.

**Table 8  Performance results on the SearchSnippets dataset using "paraphrase-distilrobertabase-v1" Pre-trained BERT model.**

| Clustering methods | SIL | NMI | ARI | FMI |
|---|---|---|---|---|
| Sentence similarity measure with embedding techniques | | | | |
| Partitional clustering ($k$-means) | 0.107 | 0.682 | 0.926 | 0.711 |
| Hierarchical clustering (Agglomerative) | 0.033 | 0.861 | 0.968 | 0.879 |
| Fuzzy clustering (Fuzzy-Relational) | 0.006 | 0.196 | 0.716 | 0.246 |
| Sentence similarity measure without embedding techniques | | | | |
| Partitional clustering ($k$-means) | 0.009 | 0.330 | 0.375 | 0.290 |
| Hierarchical clustering (Agglomerative) | 0.001 | 0.386 | 0.320 | 0.345 |
| Fuzzy clustering (Fuzzy-Relational) | −0.032 | 0.020 | 0.288 | 0.134 |

The clustering results obtained using different pre-trained BERT models and techniques shed light on the importance of model selection and embedding strategies in text clustering tasks. The superiority of the agglomerative method in both experimental scenarios suggests its robustness in forming cohesive clusters, which can be attributed to its agglomerative nature that combines similar data points progressively. Additionally, the "*paraphrase-distilroberta-base-v1*" model consistently archives improved results, indicating the significance of model architecture and training data in capturing semantic relationships between sentences, which directly influences clustering quality. Moreover, the decline in performance when embedding techniques are not employed emphasizes the role of embeddings in improving the representation of sentences and facilitating more effective clustering. These findings offer valuable guidance for practitioners and researchers in choosing the right combination of pre-trained models and clustering methods for text data, ensuring better cluster quality and alignment with ground truth labels in NLP applications.

Figures 10, 11 and 12 visually represent the clustering performance of $k$-means and agglomerative clustering methods using SentenceBERT measures on the SearchSnippets dataset. Each figure contains four subplots, two for $k$-means and two for agglomerative clustering, and they compare the performance of two pre-trained BERT models, "*bert-base-nli-mean-tokens*" and "*paraphrase-distilroberta-base-v1*". In Figs. 10A and 10B, we observe the Silhouette performance of $k$-means using "*bert-base-nli-mean-tokens*" and "*paraphrase-distilroberta-base-v1*" models, respectively. These plots provide a visual representation of cluster quality. Higher Silhouette scores indicate better separation between clusters. We can see that the "*paraphrase-distilroberta-base-v1*" model consistently outperforms "*bert-base-nli-mean-tokens*" in $k$-means clustering, indicating that the former is more effective at forming well-separated clusters. In Figs. 10C and 10D, however, the Silhouette performance of agglomerative clustering with the same pre-trained models is depicted. Similar to $k$-means, agglomerative clustering using "*paraphrase-distilroberta-base-v1*" consistently shows higher Silhouette scores compared to "*bert-base-nli-mean-tokens*". This suggests that the choice of the pre-trained BERT model has a significant impact on the cluster quality, with "*paraphrase-distilroberta-base-v1*" yielding better results.

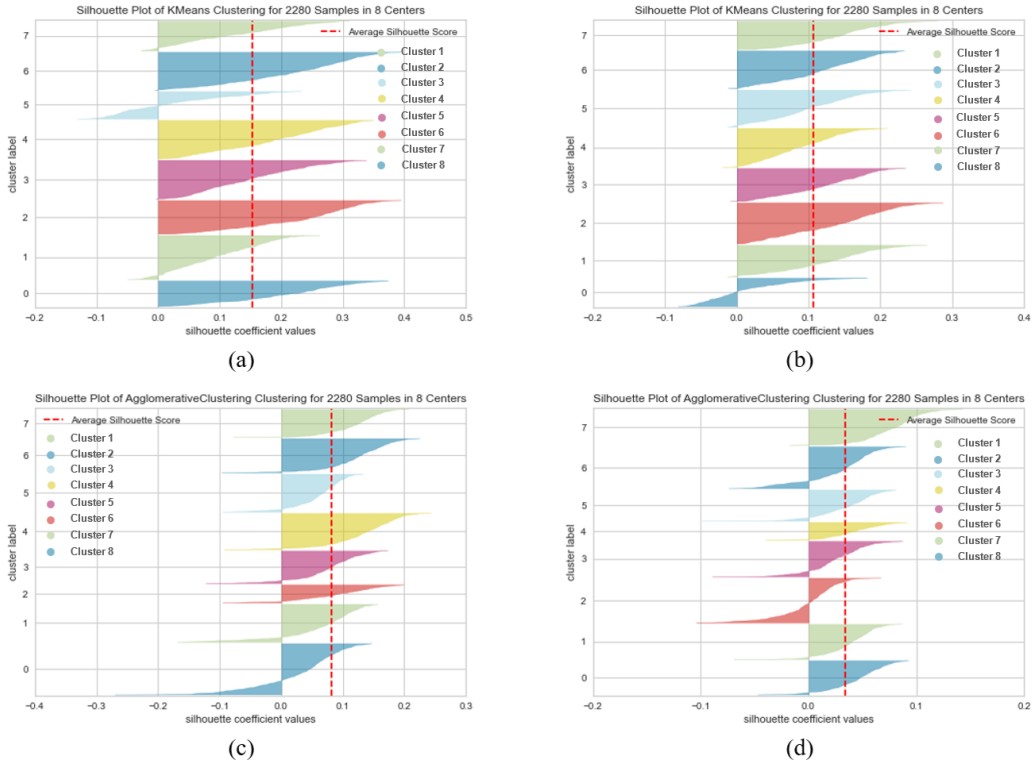

**Figure 10 K-means and agglomerative clustering results of the used SentenceBERT measure on SearchSnippets dataset, where one colour represent a cluster.** Graphs (A) and (B) show the silhouette performance of *k*-means method using pre-trained BERT models "*bert-base-nli-mean-tokens*" and "*paraphrase-distilroberta-base-v1*" respectively. Graphs (C) and (D) show the Silhouette performance of agglomerative method using pre-trained BERT models "*bert-base-nli-mean-tokens*" and "*paraphrase-distilroberta-base-v1*", respectively.

The visual analysis of clustering results in Figs. 10, 11 and 12 reaffirms the findings from the quantitative evaluation. The superiority of the "*paraphrase-distilroberta-base-v1*" model over "*bert-base-nli-mean-tokens*" is visually evident in both *k*-means and agglomerative clustering. The Silhouette scores consistently show that "*paraphrase-distilroberta-base-v1*" leads to better-defined clusters with higher inter-cluster separation and intra-cluster cohesion. Furthermore, the visual representation highlights the importance of selecting the appropriate clustering method. Both *k*-means and agglomerative clustering exhibit improved performance with "*paraphrase-distilroberta-base-v1*" but agglomerative clustering consistently outperforms *k*-means in forming well-separated clusters. This aligns with the earlier discussion, emphasizing the significance of hierarchical clustering methods in clustering tasks.

### Comparison with short text clustering approaches

In Table 9, we present a performance comparison of our best clustering results against recent short text clustering results reported in *Yin et al. (2021)* on the three datasets mentioned above. We have chosen ARI and NMI metrics to measure clustering performance. All of

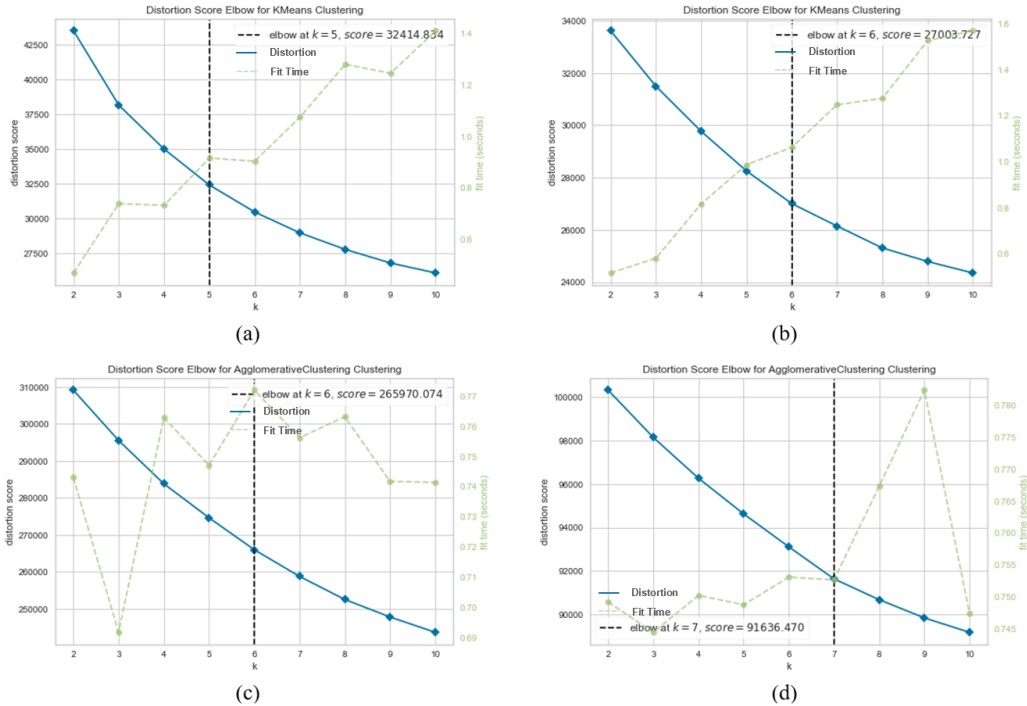

**Figure 11** *K*-means and agglomerative clustering results of the used SentenceBERT measure on SearchSnippets dataset, where one colour represent a cluster. Graphs (A) and (B) show the silhouette performance of *k*-means method using pre-trained BERT models "*bert-base-nli-mean-tokens*" and "*paraphrase-distilroberta-base-v1*" respectively. Graphs (C) and (D) show the Silhouette performance of agglomerative method using pre-trained BERT models "*bert-base-nli-mean-tokens*" and "*paraphrase-distilroberta-base-v1*", respectively.

**Table 9** Performance of our clustering results and baselines.

| Clustering method | Evaluation matric | MR dataset | AG news dataset | SearchSnippets dataset |
|---|---|---|---|---|
| BoW | ARI | 0.518 | 0.280 | 0.245 |
| | NMI | 0.003 | 0.009 | 0.089 |
| BERT | ARI | 0.784 | 0.574 | 0.667 |
| | NMI | 0.251 | 0.257 | 0.479 |
| SCA-AE | ARI | 0.766 | 0.683 | 0.687 |
| | NMI | 0.219 | 0.341 | 0.502 |
| Our Results | ARI | 0.585 | 0.740 | 0.968 |
| | NMI | 0.292 | 0.393 | 0.861 |

our best results were achieved using the agglomerative clustering method, employing the "*paraphrase-distilroberta-base-v1*" pre-trained BERT model. It is noteworthy, however, that for the MR dataset, the highest ARI score was attained using the *k*-means clustering method with the "*bert-base-nli-mean-tokens*" pre-trained BERT model.

As depicted in Table 9, the Bag of Words (BoW) method (*i.e.,* a traditional text representation approach) exhibited moderate performance on the MR dataset with an
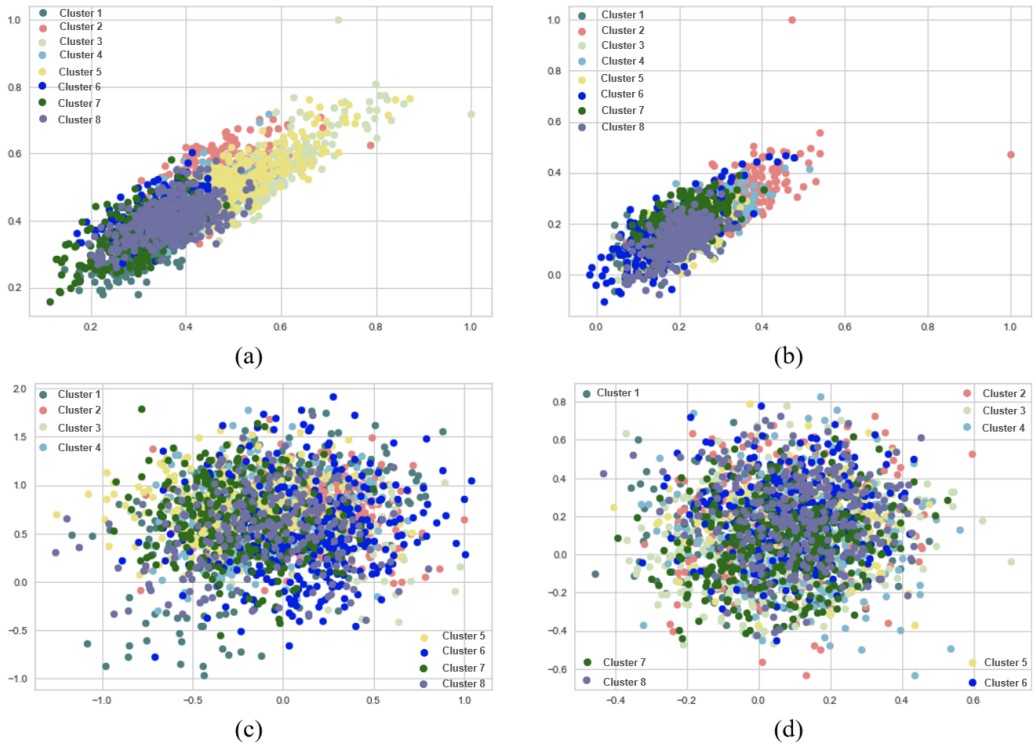

**Figure 12** ***K*.-means and agglomerative clustering results of the used SentenceBERT measure on SearchSnippets dataset, where one colour represent a cluster.** Graphs (A) and (B) show the silhouette performance of *k*-means method using pre-trained BERT models "*bert-base-nli-mean-tokens*" and "*paraphrase-distilroberta-base-v1*" respectively. Graphs (C) and (D) show the Silhouette performance of agglomerative method using pre-trained BERT models "*bert-base-nli-mean-tokens*" and "*paraphrase-distilroberta-base-v1*", respectively.

ARI score of 0.518. However, its performance significantly declined on the AG News and SearchSnippets datasets. Additionally, BoW produced low NMI scores across all datasets, underscoring its limitations in capturing meaningful text clusters. In contrast, BERT-based clustering, leveraging contextual embeddings, consistently outperformed BoW on all datasets. Notably, it achieved high ARI scores on the MR dataset (0.784) and AG News dataset (0.574), illustrating its effectiveness in capturing textual context, including semantic information. In terms of NMI, BERT-based clustering also surpassed BoW across all datasets, indicating an enhancement in cluster quality.

Sparse Coding Autoencoder (SCA-AE) demonstrated competitive performance, especially on the AG News and SearchSnippets datasets, with ARI scores of 0.683 and 0.687, respectively. It also achieved the highest NMI score among the baseline methods on the SearchSnippets dataset, signifying strong agreement between predicted clusters and actual clusters. Conversely, our clustering results consistently outperformed all other methods across all datasets. Notably, it achieved an impressive ARI score of 0.968 on the SearchSnippets dataset, indicating highly accurate clustering results. Regarding NMI, our results excelled by attaining the highest scores across all three datasets, demonstrating a

strong mutual information association with true clusters. Finally, our evaluation framework exhibits significant potential to enhance clustering performance across diverse text datasets, rendering it well-suited for applications such as information retrieval and document summarization.

## Application to text summarization

While our primary focus has been on sentence clustering as a generic activity, it often comes into play within various text-processing tasks, such as extractive document summarization (*Srivastava et al., 2023*). In extractive summarization, the goal is to select a relatively small subset of sentences for inclusion in a summary. One straightforward approach to utilize clustering results in generating an extractive summary is to pick the most centric sentence from each cluster (*i.e.,* cluster's centroid). However, depending on the number of clusters identified, this may lead to either too few or too many sentences in the summary. Therefore, we may need to consider adding or removing sentences from this summary, and several approaches are available for this purpose. For instance, if we intend to include more sentences, we can choose additional similar sentences from each cluster, but this might result in an excessively lengthy summary with potential content duplication.

To illustrate how sentence clustering can be applied in an end-to-end (*in vivo*) context, we have employed the best-performing sentence clustering algorithm in our experiment, the agglomerative clustering method (*Roux, 2018*), in conjunction with the "*paraphrase-distilroberta-base-v1*" pre-trained BERT model to summarize a recent news article. Table 10 presents the sentences from this article which discusses the use of ChatGPT in universities. We have selected this article simply because it was topical at the time of our study and represented the typical length and content breadth commonly encountered in text processing tasks like text summarization.

A more intuitive appreciation of the agglomerative clustering algorithm performance on summarizing news article can be gained by using two evaluation benchmarks: (i) cluster the news article using agglomerative clustering algorithm (here we only identify four clusters as *threshold*), and (ii) human clustering's. In the first case, sentences in the news article were clustered to four clusters and identification of one sentence as being central to each cluster (*i.e.,* most similar). Therefore, the initial summary resulting from selecting the four clusters centroid would consist of sentences 1, 10, 22 and 28. For human-assigned clustering's, we provided 10 humans with this news article and asked them to identify the four most important sentences that capture the main ideas of the article (*i.e.,* reference/ground-truth summary). The evaluation here is conducted based on human judges' scores and cosine similarity scores. The agreement between the humans' judges ranged from a minimum of 85% to a maximum of 95%, with a mean of 90%. Additionally, we computed the cosine similarity between the human-selected sentences (the reference summary) and the news article, achieving a similarity score of 85%. Table 11 presents a comparison of human-generated summaries (reference summaries) and summaries generated through agglomerative clustering.

Both the reference summary and the agglomerative clustering generated summary received a high human judges score of 90%, indicating that human judges found both

**Table 10  News article dataset broken down into sentences.**

**Title: ChatGPT: Cardiff students admit using AI on essays (*Wild, 2023*)**

Cardiff University students said they had received first class grades for essays written using the AI chatbot.

ChatGPT is an AI program capable of producing human-like responses and academic pieces of work.

Cardiff University said it was reviewing its policies and would issue new university-wide guidance shortly.

Tom, not his real name, is one of the students who conducted his own experiment using ChatGPT.

Tom, who averages a 2.1 grade, submitted two 2,500 essays in January, one with the help of the chatbot and one without.

For the essay he wrote with the help of AI, Tom received a first - the highest mark he has ever had at university.

In comparison, he received a low 2.1 on the essay he wrote without the software.

I didn't copy everything word for word, but I would prompt it with questions that gave me access to information much quicker than usual, said Tom.

He also admitted that he would most likely continue to use ChatGPT for the planning and framing of his essays.

A recent Freedom of Information request to Cardiff University revealed that during the January 2023 assessment period, there were 14,443 visits to the ChatGPT site on the university's own wi-fi networks.

One month before, there were zero recorded visits.

Despite the increase in visits during January's assessment period, the university believes there is nothing to suggest that the visits were for illegitimate purposes.

Most visits have been identified as coming from our research network - our School of Computer Science and Informatics, for example, has an academic interest in the research and teaching of artificial intelligence, said Cardiff University.

John, not his real name, is another student at the university who admitted using the software to help him with assignments.

I've used it quite a few times since December. I think I've used it at least a little bit for every assessment I've had, he said.

It's basically just become part of my work process, and will probably continue to be until I can't access it anymore.

When I first started using it, I asked it to write stuff like *compare this niche theory with this other niche theory in an academic way* and it just aced it.

Although ChatGPT does not insert references, John said he had no issue filling those in himself.

I've also used it to summarise concepts from my course that I don't think the lecturers have been great at explaining, he said.

It's a really good tool for cutting out the waffle that some lecturers go into for theories which you don't actually need to talk about in essays.

It probably cuts about 20% of the effort I would need to put into an essay.

Both students said they do not use ChatGPT to write their essays, but to generate content they can tweak and adapt themselves.

As for being caught, John is certain that the AI influence in his work is undetectable.

I see no way that anyone could distinguish between work completely my own and work which was aided by AI, he said.

However, John is concerned about being caught in the future. He said if transcripts of his communication with the AI network were ever found, he fears his degree could be taken away from him.

I'm glad I used it when I did, in the final year of my degree, because I feel like a big change is coming to universities when it comes to coursework because it's way too easy to cheat with the help of AI, he said.

**Table 10** (*continued*)

| Title: ChatGPT: Cardiff students admit using AI on essays (*Wild, 2023*) |
| --- |
| I like to think that I've avoided this, whilst reaping the benefits of GPT in my most important year. |
| Cardiff University said it took allegations of academic misconduct, including plagiarism, extremely seriously. |
| Although not specifically referenced, the improper use of AI would be covered by our existing academic integrity policy, a spokesman said. |
| We are aware of the potential impact of AI programmes, like ChatGPT, on our assessments and coursework. |
| Maintaining academic integrity is our main priority and we actively discourage any student from academic misconduct in its many forms. |

**Table 11** Human clustering *vs.* agglomerative algorithm clustering performance on news article dataset.

| | Article summary |
| --- | --- |
| **Human generated summary** (reference summary) | 1. Cardiff University students said they had received first class grades for essays written using the AI chatbot. |
| | 10. A recent Freedom of Information request to Cardiff University revealed that during the January 2023 assessment period, there were 14,443 visits to the ChatGPT site on the university's own wi-fi networks. |
| | 22. Both students said they do not use ChatGPT to write their essays, but to generate content they can tweak and adapt themselves. |
| | 28. Cardiff University said it took allegations of academic misconduct, including plagiarism, extremely seriously. |
| **Agglomerative clustering generated summary** | 1. Cardiff University students said they had received first class grades for essays written using the AI chatbot |
| | 22. Both students said they do not use ChatGPT to write their essays, but to generate content they can tweak and adapt themselves |
| | 28. Cardiff University said it took allegations of academic misconduct, including plagiarism, extremely seriously |
| | 10. A recent Freedom of Information request to Cardiff University revealed that during the January 2023 assessment period, there were 14,443 visits to the ChatGPT site on the university's own wi-fi networks. |

summaries to be of similar quality. This suggests that the agglomerative clustering approach effectively generates summaries that are on par with human-authored ones. The cosine similarity score of 85% further supports this finding, indicating substantial content overlap between the reference and agglomerative clustering generated summaries. While these results are promising, it is important to note that the agglomerative clustering method does not account for linguistic fluency, coherence, or style, which are attributes that human authors naturally incorporate into their summaries. Therefore, there may still be room for improvement in generating more human-like summaries. Nevertheless, the study highlights the potential scalability and efficiency of using automated methods like agglomerative clustering for summarization tasks, particularly when dealing with large datasets where manual summarization would be time-consuming and resource-intensive. These findings have significant implications for the field of text summarization, potentially revolutionizing the way we handle and extract information from extensive textual data in various applications.

In addition to human judges' scores and cosine similarity scores, we employed standard summarization evaluation metrics such as ROUGE-1, ROUGE-2 and ROUGE-L (*Lin &*

*Hovy, 2003*; *Lin, 2004*) to assess the quality of the extractive summary generated through the agglomerative clustering approach. The results, as shown in Fig. 13, provide valuable insights into the performance of the summarization method. Firstly, the ROUGE-1 scores for precision, recall, and F1 scores are all exceptionally high, each registering a perfect score of 1.00. This indicates that the extractive summary precisely captured every unigram from the reference summary, reflecting a flawless level of overlap and agreement between the generated summary and the reference summary at the word level. Secondly, the ROUGE-2 scores for precision, recall, and F1 score are notably high, with each metric achieving a score of 0.97. ROUGE-2 evaluates the overlap between bigrams in the generated summary and the reference summary. The high scores here suggest that the agglomerative clustering approach excelled in capturing significant word pairs, further demonstrating its ability to retain essential content from the source text. However, it is important to note that the ROUGE-L scores for precision, recall, and F1 scores are comparatively lower, each registering at 0.60. ROUGE-L evaluates the longest common subsequence between the generated summary and the reference summary. The lower scores in this case may be attributed to differences in sentence structure, ordering, or variations in wording between the two summaries.

The summarization results depicted in Fig. 13, particularly ROUGE-1 and ROUGE-2, provide strong evidence of the agglomerative clustering approach's effectiveness in producing extractive summaries that align closely with the reference summary at both the unigram and bigram levels. However, the comparatively lower ROUGE-L scores indicate that there may still be room for improvement in terms of capturing the longest common subsequence between the generated and reference summaries, possibly by considering sentence restructuring or reordering. Nevertheless, the overall high performance of the summarization method, as indicated by these metrics, underscores its potential as a valuable tool for automated text summarization tasks.

## DISCUSSION

The developed evaluation framework presented above serves as a pivotal cornerstone for advancing downstream natural language processing applications, particularly in realms such as information retrieval, optimization processes, and document summarization. The limitations of word-level embeddings in capturing nuanced semantic information at the sentence level have been a significant impediment to the efficacy of various text-processing tasks. Our framework, centered on embedding-based semantic similarity measures for evaluating unsupervised clustering methods at the sentence level, addresses this crucial gap. This innovation is pertinent to a diverse audience involved in natural language processing research and applications, including data scientists, NLP practitioners, and researchers seeking to improve the performance of text-related tasks. Moreover, managers and decision-makers within organizations leveraging NLP technologies for information retrieval, content summarization, or optimization processes stand to benefit significantly from our research results. The demonstrated efficacy of incorporating sentence embedding techniques into unsupervised clustering methodologies offers valuable insights
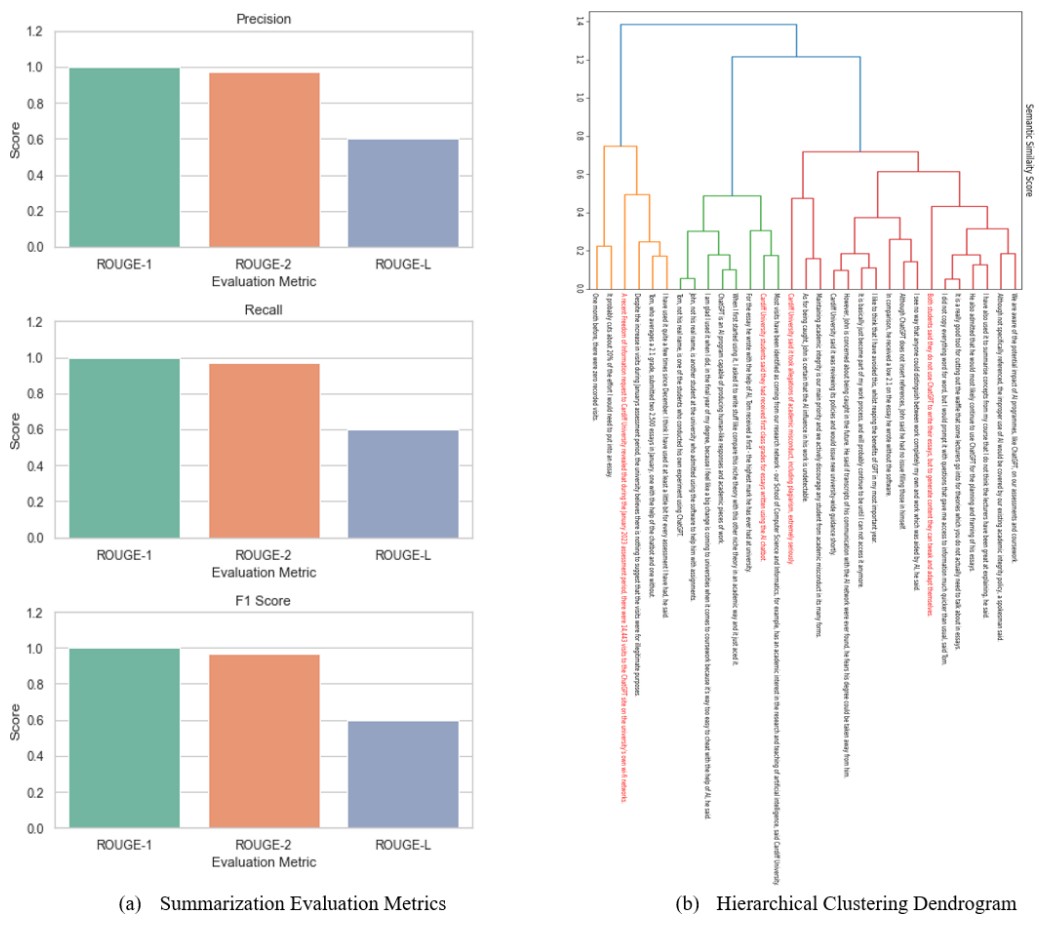

(a)  Summarization Evaluation Metrics          (b)  Hierarchical Clustering Dendrogram

**Figure 13  Performance of agglomerative clustering algorithm on summarizing a news article.** Graph (A) shows the standard evaluation metrics for text summarization. Graphs (B) shows the most important sentences identified by agglomerative clustering.

and practical guidance. Managers can use these findings to enhance the accuracy and efficiency of information retrieval systems, streamline document summarization processes, and refine optimization strategies in various business domains. For these stakeholders, our suggestion is to consider integrating embedding-based techniques into their existing NLP pipelines or systems to augment the performance of text-related tasks. Additionally, investing in further research and development focused on refining embedding models, exploring hybrid approaches, and adapting the framework to domain-specific applications would be instrumental in harnessing the full potential of embedding-based methods for improved text processing outcomes.

## CONCLUSION

This study has addressed a critical gap in the evaluation of sentence-level text clustering methods by introducing and implementing a novel approach that leverages embedding-based semantic similarity measures. While sentence embedding measures have gained

popularity in computational linguistics, their application to evaluate unsupervised clustering methods for sentence-level text has been limited. This study, driven by the hypothesis that integrating sentence embedding techniques could improve the efficacy of these clustering methods, offers significant contributions.

First and foremost, this study represents the pioneering effort to establish an experimental and evaluation framework enabling the comparison of various unsupervised clustering methods using sentence embedding techniques trained on state-of-the-art pre-training models. We have meticulously examined and evaluated the performance of three distinct clustering approaches: partitional clustering, hierarchical clustering, and fuzzy clustering, all of which utilize the SentenceBERT sentence similarity measure. This measure, based on vector embeddings, serves as the foundation for text representation. Additionally, we have explored the capabilities of this measure by training it with two different pre-training models, striving to capture richer and more accurate semantic information.

Our comparative analysis within this framework showcased that, while traditional methods like Bag-of-Words (BoW) achieved moderate performance, BERT-based clustering significantly outshone BoW, demonstrating its superior capability in capturing semantic nuances across different datasets. Similarly, SCA-AE displayed competitive performance, particularly on the AG News and SearchSnippets datasets, with our proposed methods surpassing all others in accuracy and cluster quality as indicated by ARI and NMI scores. These findings underscore the effectiveness of employing sentence embedding similarity measures in short text clustering tasks and highlight our novel approach's significant contribution to text summarization and information retrieval applications.

Furthermore, our work goes beyond theoretical exploration by conducting extensive experiments on large-scale benchmark datasets. The results unequivocally support the notion that incorporating sentence embedding techniques leads to favorable performance outcomes in both text clustering and text summarization tasks.

While this study marks a significant step in demonstrating the potential of sentence embedding techniques for enhancing unsupervised clustering methods, several avenues for future research emerge from our findings. One potential direction involves investigating the adaptability and generalizability of the proposed framework to diverse domains beyond textual data, such as multimedia content analysis or domain-specific language processing. Additionally, exploring hybrid approaches that combine embedding-based measures with domain-specific knowledge or context-aware models could further refine the performance of clustering methods in specialized applications. Furthermore, advancing the sophistication of pre-training models for generating sentence embeddings and investigating the impact of different training strategies on clustering performance remains an area ripe for exploration. Finally, extending the evaluation framework to encompass dynamic and evolving datasets could provide insights into the adaptability of clustering methods in real-time scenarios.

In summary, this research fills a significant void in the field of sentence-level text clustering and provides valuable insights into the practical benefits of integrating embedding-based semantic similarity measures into unsupervised clustering

methodologies. These findings have the potential to substantially enhance the performance of text clustering and summarization tasks, offering promising avenues for future research and application in various text-processing domains.

### Funding

The research leading to these results has received funding from the Mohammed Bin Rashid Smart Learning Program, UAE, under the Funding Agreement No: MBRSLP/02/23. The funders had no role in study design, data collection and analysis, decision to publish, or preparation of the manuscript.

### Grant Disclosures

The following grant information was disclosed by the authors:
The Mohammed Bin Rashid Smart Learning Program, UAE: MBRSLP/02/23.

### Competing Interests

The authors declare there are no competing interests.

### Author Contributions

- Khaled Abdalgader conceived and designed the experiments, performed the experiments, analyzed the data, performed the computation work, prepared figures and/or tables, authored or reviewed drafts of the article, and approved the final draft.
- Atheer A. Matroud analyzed the data, performed the computation work, prepared figures and/or tables, authored or reviewed drafts of the article, and approved the final draft.
- Khaled Hossin analyzed the data, prepared figures and/or tables, authored or reviewed drafts of the article, and approved the final draft.

### Data Availability

The MR-dataset is available at Kaggle:

https://www.kaggle.com/datasets/zhengcoming/mr-dataset/

The AG News Dataset is available at Kaggle:

https://www.kaggle.com/datasets/amananandrai/ag-news-classification-dataset

The Searchsnippets Dataset is available at Kaggle:

https://www.kaggle.com/datasets/liudad/searchsnippets

The code is available at GitHub and Zenodo:

- https://github.com/KhaledBalawafi/Transformer-based-Semantic-Similarity-Measure-for-Text-Clustering.git

- Abdalgader, K. (2024). Experimental Study on Short-text Clustering Using Transformer-based Semantic Similarity Measure. Zenodo. https://doi.org/10.5281/zenodo.10951584

## Supplemental Information

Supplemental information for this article can be found online at http://dx.doi.org/10.7717/peerj-cs.2078#supplemental-information.

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
