# Peer review of "Experimental study on short-text clustering using transformer-based semantic similarity measure"

_PeerJ Computer Science, doi:10.7717/peerj-cs.2078_

## Round 0.1 · original submission · Major Revisions

The work is interesting and has value. However, the reviewers also have some concerns about originality, experiments, etc. The authors need to improve the work according to the comments.

Reviewer 1 ·

Basic reporting

Clear and unambiguous professional English is used throughout the paper, despite the hard use of numerous acronyms that make the reading laborious.
Literature references are not sufficient: in particular, when dealing with literature on semantic-based measures with an external knowledge base, only ontologies and semantic networks are considered, while web-based measures are ignored. In fact, there are several lines of research on web-based semantics applied to sentence similarity and concept proximity, not only for the detection of text proximity in meaning (with sentence comparison or heuristic semantic walk in web-based concept networks like Wikipedia), but also for semantic similarity of sentences, with application like emotion recognition (with emotion vector extraction based on a model like Ekman or Plutchick, or semantic model for emotion recognition in web objects, e. g. comments on a topic in any web repository or social/communication network), the context-based semantic similarity of images using image recognition from metadata, context recognition in social networks.
Word embedding is also an approach often used for text mining and sentiment analysis or emotion/topic recognition. The real novelty of this paper lies in the use of word embedding together to be applied to deep learning classification using transformers, which is the latest state-of-the-art, with clustering.
In the absence of some state-of-the-art analysis, the comparison of results with existing approaches and methods is lacking, and it is not clear if the proposed methodology can improve the literature, and with what limits and comparison.
There is sufficient clarity and expressiveness in the article structure, figures, tables, and data.

Experimental design

Motivations for the choice of the datasets should be provided, besides being popular for general applications for topic classification. Similar more detailed motivations should be given for the choice of k-means and agglomerative clustering, with no refinement with respect to the models provided by recent research on clustering.

It is stated how research fills an identified knowledge gap, discarding web-based approaches to sentence-based semantics, thus weakening the motivations.
Methods are described with sufficient detail to replicate the approach, it would be beneficial to have access to the whole code and preprocessed data. In fact, only a few main functions are available in the GitHub repository, with no readme file or instructions for reproducing the results.

Validity of the findings

The claim that “This study, driven by the hypothesis that integrating 451 sentence embedding techniques could improve the efficacy of these clustering methods, offers significant 452 contributions.” Should be from a reviewer more than from the author.
The conclusions speak more of the goodness of the paper than the goodness of the results, which should include detailed analysis and comparison of results with the state of the art.

Reviewer 2 ·

Basic reporting

The abstract provides a clear overview of the research, addressing the importance of sentence clustering and highlighting the gap in evaluating clustering performance using low-dimensional continuous representations. It effectively introduces the new implementation incorporating a sentence similarity measure based on embedding representation.

The author is encouraged to incorporate recent references to enhance the incorporation of the latest developments in the field and improve the coverage of the state-of-the-art literature.

The overall structural organization of the article is satisfactory.

The figures and tables have been appropriately presented and labeled.

It would be beneficial for the author to include information about the improvement in results or accuracy in the abstract for a more comprehensive overview.

Experimental design

no comment

Validity of the findings

The conclusion would be strengthened by including a discussion on the achieved accuracy and results, providing a more comprehensive overview. It is recommended that the authors explicitly address this aspect in the conclusion section.

Reviewer 3 ·

Basic reporting

This paper is well written in English and it is clear.
The reference is sufficient.
The organization of this paper is professional.
It is also self-contained.
Most of the formal results are included.

Experimental design

By experiments, authors try to demonstates that incorporating the sentence embedding measure leads to significantly improved performance in text clustering tasks. This sounds interesting. And the experiments are described detaily and informative.

However, the experimenatal study was not supported by theoretical analysis in this paper and it is not enough to give the conclusion.

Validity of the findings

I don't think the originality in this paper is enough. The author should carefully polished the paper and try to clearly provid the main idea and the theoretical supports.

---

## Round 0.2 · accepted · Accept

The authors have revised the paper accordingly. The reviewer is satisfied with this version. It can be accepted now.

Reviewer 3 ·

Basic reporting

This paper has been revised better than the last version.

Experimental design

no comment

Validity of the findings

The findings are well organized in this version.